



**The new Kr-86 excess ice core proxy for synoptic activity: West Antarctic**
**storminess possibly linked to ITCZ movement through the last deglaciation**
Christo Buizert[1], Sarah Shackleton[2], Jeffrey P. Severinghaus[2], William H. G. Roberts[3], Alan Seltzer[2,4],
Bernhard Bereiter[5], Kenji Kawamura[6], Daniel Baggenstos[5], Anaïs J. Orsi[7,8], Ikumi Oyabu[6],
Benjamin Birner[2], Jacob D. Morgan[2], Edward J. Brook[1], David M. Etheridge[9,10], David Thornton[9],
Nancy Bertler[11,12], Rebecca L. Pyne[11], Robert Mulvaney[13], Ellen Mosley-Thompson[14], Peter D. Neff[15,16],
and Vasilii V. Petrenko[16]
[1]College of Earth, Ocean and Atmospheric Sciences, Oregon State University, Corvallis, OR 97331, USA
[2]Scripps Institution of Oceanography, University of California San Diego, La Jolla, CA 92093, USA
[3]Geography and Environmental Sciences, Northumbria University, Newcastle, UK and BRIDGE, School
of Geographical Sciences, University of Bristol, Bristol, UK
[4]Marine Chemistry and Geochemistry Department, Woods Hole Oceanographic Institution, Woods Hole,
MA 02543, USA
[5]Climate and Environmental Physics, Physics Institute, and Oeschger Center for Climate Research,
University of Bern, 3012, Bern, Switzerland
[6]National Institute for Polar Research, 10-3 Midori-cho, Tachikawa, Tokyo 190-8518, Japan
[7]Laboratoire des Sciences du Climat et de l'Environnement, LSCE/IPSL, CEA-CNRS-UVSQ, Université
Paris-Saclay,l'Orme des merisiers, Gif-sur-Yvette, France
[8]Earth, Ocean and Atmospheric Sciences Department, The University of British Columbia, Vancouver, BC
V6T 1Z4, Canada
[9]CSIRO Oceans and Atmosphere, PMB 1, Aspendale, Victoria 3195, Australia
[10]Australian Antarctic Program Partnership, Institute for Marine & Antarctic Studies, University of
Tasmania, Hobart, Tasmania 7004, Australia
[11]Antarctic Research Centre, Victoria University of Wellington, Wellington, 6012, New Zealand
[12]GNS Science, Lower Hut 5010, New Zealand
[13]British Antarctic Survey, National Environment Research Council, Cambridge CB3 0ET, UK
[14]Byrd Polar and Climate Research Center, The Ohio State University, Columbus, OH 43210, USA
[15]Department of Soil, Water, and Climate, University of Minnesota, Saint Paul, MN 55108, USA
[16]Department of Earth and Environmental Sciences, University of Rochester, Rochester, NY 14627, USA
*Correspondence to*: Christo Buizert (christo.buizert@oregonstate.edu)





**Abstract**
Here we present a newly developed ice core gas-phase proxy that directly samples a component of the
large-scale atmospheric circulation: synoptic-scale pressure variability. Surface pressure variability weakly
disrupts gravitational isotopic settling in the firn layer, which is recorded in krypton-86 excess ($^{86}Kr_{xs}$). We
validate $^{86}Kr_{xs}$ using late Holocene ice samples from eleven Antarctic and one Greenland ice core that
collectively represent a wide range of surface pressure variability in the modern climate. We find a strong
correlation ($r = -0.94$, $p < 0.01$) between site-average $^{86}Kr_{xs}$ and site synoptic variability from reanalysis
data. The main uncertainties in the method are the corrections for gas loss and thermal fractionation, and
the relatively large scatter in the data. We show $^{86}Kr_{xs}$ is linked to the position of the eddy-driven subpolar
jet (SPJ), with a southern position enhancing pressure variability.
We present a $^{86}Kr_{xs}$ record covering the last 24 ka from the WAIS Divide ice core. West Antarctic synoptic
activity is slightly below modern levels during the last glacial maximum (LGM); increases during the
Heinrich Stadial 1 and Younger Dryas North Atlantic cold periods; weakens abruptly at the Holocene onset;
remains low during the early and mid-Holocene, and gradually increases to its modern value. The WAIS
Divide $^{86}Kr_{xs}$ record resembles records of monsoon intensity thought to reflect changes in the meridional
position of the intertropical convergence zone (ITCZ) on orbital and millennial timescales, such that West
Antarctic storminess is weaker when the ITCZ is displaced northward, and stronger when it is displaced
southward. We interpret variations in synoptic activity as reflecting movement of the South Pacific SPJ in
parallel to the ITCZ migrations, which is the expected zonal-mean response of the eddy-driven jet in models
and proxy data. Past changes to Pacific climate and the El Niño Southern Oscillation (ENSO) may amplify
the signal of the SPJ migration. Our interpretation is broadly consistent with opal flux records from the
Pacific Antarctic zone thought to reflect wind-driven upwelling.
We emphasize that $^{86}Kr_{xs}$ is a new proxy, and more work is called for to confirm, replicate and better
understand these results; until such time, our conclusions regarding past atmospheric dynamics remain
tentative. Current scientific understanding of firn air transport and trapping is insufficient to explain all the
observed variations in $^{86}Kr_{xs}$.



# 1 Introduction

## 1.1 Motivation and objectives

Proxy records from around the globe show strong evidence for past changes in Earth's atmospheric circulation and hydrological cycle that often far exceed those seen in the relatively short instrumental period.

For example, low-latitude records of riverine discharge captured in ocean sediments (Peterson et al., 2000), and isotopic composition of meteoric water captured in dripstone calcite (Cheng et al., 2016), suggest large variations in tropical hydrology and monsoon strength, commonly interpreted as meridional migrations of the intertropical convergence zone or ITCZ (Chiang and Friedman, 2012; Schneider et al., 2014). Such ITCZ movement is seen both in response to insolation changes linked to planetary orbit (Cruz et al., 2005) as well as in response to the abrupt millennial-scale Dansgaard-Oeschger (D-O) and Heinrich cycles of the North-Atlantic (Kanner et al., 2012; Wang et al., 2001); the organizing principle is that the ITCZ follows the thermal equator and therefore migrates towards the warmer (or warming) hemisphere (Broccoli et al., 2006; Chiang and Bitz, 2005).

As a second example, the intensity of the El Niño – Southern Oscillation (ENSO), the dominant mode of global interannual climate variability, has changed through time. A variety of proxy data suggest ENSO activity in the 20th century was much stronger than in preceding centuries (Emile-Geay et al., 2015; Fowler et al., 2012; Gergis and Fowler, 2009; Thompson et al., 2013). The vast majority of data and model studies suggest weakened ENSO strength in the mid- and early-Holocene, likely in response to stronger orbitally-driven NH summer insolation at that time (Braconnot et al., 2012; Cane, 2005; Clement et al., 2000; Driscoll et al., 2014; Koutavas et al., 2006; Liu et al., 2000; Liu et al., 2014; Moy et al., 2002; Rein et al., 2005; Tudhope et al., 2001; Zheng et al., 2008); yet other studies suggest there may not be such a clear trend, and simply more variability (Cobb et al., 2013). Intensification of ENSO (or perhaps a more El-Niño-like mean state) may have occurred during the North-Atlantic cold phases of the abrupt D-O and Heinrich cycles (Braconnot et al., 2012; Merkel et al., 2010; Stott et al., 2002; Timmermann et al., 2007). Overall, understanding past and future ENSO variability remains extremely challenging (Cai et al., 2015).

As a last example, the strength and meridional position of the southern hemisphere westerlies (SHW) is thought to have changed in the past, which, via Southern Ocean wind-driven upwelling, has potential implications for the global overturning circulation (Marshall and Speer, 2012) and for carbon storage in the abyssal ocean (Anderson et al., 2009; Russell et al., 2006; Toggweiler et al., 2006). The SHW are thought to be shifted equatorward (Kohfeld et al., 2013) during the last glacial maximum (LGM), a shift on which climate models disagree (Rojas et al., 2009; Sime et al., 2013). During the abrupt D-O and Heinrich cycles, the SHW move in parallel with the aforementioned migrations of the ITCZ in both data (Buizert et al., 2018; Marino et al., 2013; Markle et al., 2017) and models (Lee et al., 2011; Pedro et al., 2018; Rind et al., 2001).

As these examples clearly illustrate, evidence of past changes to the large-scale atmospheric circulation is widespread. However, proxy evidence of such past changes is typically indirect – for example via isotopes in precipitation, sea surface temperature, ocean frontal positions, windblown dust, or ocean upwelling – complicating their interpretation. Here we present a newly developed noble gas-based ice core proxy, Kr-





86 excess ($^{86}Kr_{xs}$), that directly samples a component of the large-scale atmospheric circulation: synoptic-
scale pressure variability. Owing to the firn air residence time of several years (Buizert et al., 2013) and the
gradual bubble trapping process, each ice core sample contains a distribution of gas ages, rather than a
single age. Therefore, $^{86}Kr_{xs}$ does not record the passing of individual weather systems, but rather the time-
average intensity of synoptic-scale barometric variability.
Here we provide the first complete description of this new proxy. We validate and calibrate $^{86}Kr_{xs}$ using
late-Holocene ice core samples from locations around Antarctica and Greenland that represent a wide range
of pressure variability in the modern climate. We discuss the difficulties in using this proxy (analytical
precision, surface melt, corrections for sample gas loss and thermal fractionation). Next, we use reanalysis
data to better understand the drivers of surface pressure variability in Antarctica. Last, we present an $^{86}Kr_{xs}$
records from the Antarctic WAIS Divide ice core through the last deglaciation.

### 1.2 Gravitational disequilibrium and Kr-86 excess

The upper 50-100 m of the ice sheet accumulation zone consists of firn, the unconsolidated intermediate
stage between snow and ice. An interconnected pore network exists within the firn, in which gas transport
is dominated by molecular diffusion (Schwander et al., 1993). Diffusion in this stagnant air column results
in gravitational enrichment in heavy gas isotopic ratios such as $\delta^{15/14}N-N_2$, $\delta^{40/36}Ar$ and $\delta^{86/82}Kr$ (Schwander,
1989; Sowers et al., 1992). In gravitational equilibrium, all these gases attain the same degree of isotopic
enrichment per unit mass difference:

$$\delta_{\text{grav}}(z) = \left[\exp\left(\frac{\Delta m g z}{RT}\right) - 1\right] \times 1000‰ \qquad (1)$$

with Δm the isotopic mass difference ($1\times10^{-3}$ kg mol$^{-1}$), $g$ the gravitational acceleration, $z$ the depth, $R$ the
gas constant and $T$ the Kelvin temperature.
Besides molecular diffusion, firn air is mixed and transported via three other processes: downward
advection with the sinking ice matrix, convective mixing (used in the firn air literature as an umbrella term
to denote vigorous air exchange with the atmosphere via e.g. wind pumping and seasonal convection), and
dispersive mixing. These last three transport processes are all driven by large-scale air movement that does
not distinguish between isotopologues, and we refer to them collectively as macroscopic air movement. Of
particular interest for our proxy is dispersive mixing, which is driven by surface pressure variations. When
a low-pressure (high-pressure) system moves into the site, firn air at all depth levels is forced upwards
(downwards) to reach hydrostatic equilibrium with the atmosphere – a process called barometric pumping.
One can think of the firn layer "breathing" in and out in response to a rising and falling barometer,
respectively. Because firn has a finite dispersivity (Schwander et al., 1988), this air movement mixes the
interstitial firn air (Buizert and Severinghaus, 2016).
Any type of macroscopic air movement disturbs the gravitational settling, reducing isotopic enrichment
below $\delta_{\text{grav}}$. Let $\delta^{86}Kr$, $\delta^{40}Ar$, and $\delta^{15}N$ refer to deviations of $^{86}Kr/^{82}Kr$, $^{40}Ar/^{36}Ar$, and $^{29}N_2/^{28}N_2$, respectively,
from their ratios in the well-mixed atmosphere. Gases that diffuse faster (such as $N_2$) will always be closer
to gravitational equilibrium than gases that diffuse slower (such as Kr), and in the absence of thermal
fractionation $\delta^{86}Kr/4 < \delta^{40}Ar/4 < \delta^{15}N < \delta_{\text{grav}}$. The isotopic differences $\delta^{86}Kr/4 - \delta^{40}Ar/4$ and $\delta^{86}Kr/4 - \delta^{15}N$
thus reflect the degree of gravitational disequilibrium. The magnitudes of the isotopic disequilibria scale in

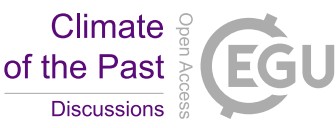

a predictable way following the molecular diffusion coefficients (Birner et al., 2018); because the diffusion coefficients of $N_2$ and Ar are very similar, their disequilibria are comparable in magnitude. We define Kr-86 excess using the Kr and Ar isotopic difference:

$$^{86}Kr_{xs40} = \frac{\delta^{86}Kr_{corr} - \delta^{40}Ar_{corr}}{\delta^{40}Ar_{corr}} \times 1000 \text{ per meg ‰}^{-1} \qquad (2)$$

where the "corr" subscript denotes a correction for gas loss (Appendix A1) and thermal fractionation (Appendix A2). The rationale for including a normalization in the denominator is discussed below. An alternative Kr-86 excess definition is possible using $\delta^{15}N$ instead of $\delta^{40}Ar$:

$$^{86}Kr_{xs15} = \frac{\delta^{86}Kr_{corr}/4 - \delta^{15}N_{corr}}{\delta^{15}N_{corr}} \times 1000 \text{ per meg ‰}^{-1} \qquad (3)$$

The $^{86}Kr_{xs40}$ definition is preferred, because it is less sensitive to thermal fractionation making it more suitable for interpreting time series. Unless explicitly stated otherwise, we use $^{86}Kr_{xs40}$ as our definition of Kr-86 excess. The $^{86}Kr_{xs15}$ does provide a way to check the validity of $^{86}Kr_{xs40}$ timeseries, and indeed we find good correspondence between both definitions for the WDC deglacial timeseries (Fig. 6). Because the disequilibrium signal is small, we express $^{86}Kr_{xs}$ in units of per meg (parts per million) of gravitational disequilibrium per ‰ of gravitational enrichment.

In the (theoretical) case of full gravitational equilibrium (and no gas loss or thermal fractionation), $\delta^{86}Kr/4$ = $\delta^{40}Ar/4 = \delta^{15}N = \delta_{grav}$, and therefore $^{86}Kr_{xs} = 0$. Any type of macroscopic mixing will cause $\delta^{86}Kr/4 <$ $\delta^{40}Ar/4 < \delta^{15}N < \delta_{grav}$, and thus $^{86}Kr_{xs} < 0$. In this sense $^{86}Kr_{xs}$ is a quantitative measure for the degree of gravitational disequilibrium in the firn layer (Birner et al., 2018; Buizert and Severinghaus, 2016).

Kawamura et al. (2013) first describe this gravitational disequilibrium (or kinetic) fractionation effect at the Megadunes site (Severinghaus et al., 2010), where deep firn cracking leads to a 23 m-thick convective zone. They suggest that the isotopic disequilibrium can be used to estimate past convective zone thickness. We show here that sites with small convective zones can nevertheless have very negative $^{86}Kr_{xs}$, and instead we suggest that the ice core $^{86}Kr_{xs}$ is dominated by dispersive mixing driven by barometric pumping from synoptic-scale pressure variability.

The principle behind $^{86}Kr_{xs}$ is illustrated with idealized firn model experiments in Fig. 1. In the absence of dispersive mixing (Fig. 1A, left panel), all isotope ratios approach $\delta_{grav}$ and $\delta^{86}Kr - \delta^{40}Ar$ is close to zero – but not exactly zero owing to downward air advection. Next, we replace a fraction $f$ of the molecular diffusion with dispersive mixing. With dispersive mixing at $f = 0.1$ and $f = 0.2$ of total mixing (middle and right panels, respectively), isotopic enrichment is progressively reduced below $\delta_{grav}$ (dashed line), making $\delta^{86}Kr - \delta^{40}Ar$ (and consequently $^{86}Kr_{xs}$) increasingly negative.

The ratio of macroscopic over diffusive transport is expressed via the dimensionless Péclet number, given here for advection and dispersion:

$$Pe_X = \frac{w_{air}L + D_{disp}}{D_X} \qquad (4)$$



where $\mathbf{Pe}_X$ is the Péclet number for gas $X$, $w_{air}$ the (downward) advective air velocity, $L$ a characteristic
length scale, $D_X$ the diffusion coefficient for gas $X$, and $D_{disp}$ is the dispersion coefficient (Buizert and
Severinghaus, 2016). In agreement with earlier studies (Birner et al., 2018; Kawamura et al., 2013), we find
that $\delta^{86}Kr$ - $\delta^{40}Ar$ is maximized when molecular and dispersive mixing are equal in magnitude ($f = 0.5$, Fig.
1B), corresponding to $\mathbf{Pe}_X \approx 1$. Note that $^{86}Kr_{xs}$ responds more linearly to $f$ than $\delta^{86}Kr$ - $\delta^{40}Ar$ does, due to
$\delta^{40}Ar$ in the denominator of Eq. (2).
In a last idealized experiment, we keep the fraction of dispersion fixed at $f = 0.1$ while we reduce the
thickness of the firn column by increasing the site temperature (Fig. 1C). We find that $\delta^{86}Kr$ - $\delta^{40}Ar$ scales
linearly with firn thickness, here represented by $\delta^{40}Ar$ on the x-axis. However, $^{86}Kr_{xs}$ remains essentially
constant due to the normalization by $\delta^{40}Ar$ in the denominator of Eq. (2). The normalization step is thus
necessary to enable meaningful comparison between different sites and time periods that all have different
firn thicknesses. For this reason, the definition of $^{86}Kr_{xs}$ used here has been updated from the original
definition by (Buizert and Severinghaus, 2016).
Note that these highly idealized experiments assume dispersive mixing to be a fixed fraction of total
transport throughout the firn column, equivalent to a constant Péclet number in the diffusive zone (a
convective zone is absent in these simulations). In reality, the Péclet number varies greatly on all spatial
scales. On the macroscopic scale (> 1 m), Pe reflects the various transport regimes (Sowers et al., 1992),
being highest in the convective and lock-in zones. On the microscopic scale (< 1 cm), hydraulic
conductance scales as $\propto r^4$ (with $r$ the pore radius) whereas the diffusive conductance scales as $\propto r^2$. This
means that the Darcy flow associated with barometric pumping will concentrate in the widest pores and
pathways, leading to a range of effective Péclet numbers within a single sample of firn. At intermediate
spatial scales of a few cm, firn density layering introduces strong heterogeneity in transport properties. It is
unclear at present whether the competition between diffusive and non-diffusive transport, which occurs at
the microscopic pore level, can be accurately represented in macroscopic firn air models via a linear
parameterization as is the current practice.



## 2 Methods

### 2.1 Ice core sites

In this study we use ice samples from eleven ice cores drilled in Antarctica, and one in Greenland. The Antarctic sites are: West Antarctic Ice Sheet (WAIS) Divide core (WDC06A, or WDC), Siple Dome (SDM), James Ross Island (JRI), Bruce Plateau (BP), Law Dome DE08, Law Dome DE08-OH, Law Dome DSSW20K, Roosevelt Island Climate Evolution (RICE), Dome Fuji (DF), EPICA (European Project for Ice Coring in Antarctica) Dome C (EDC), and South Pole Ice Core (SPC14, or SP). Ice core locations in Antarctica are shown in Fig. 2A. In Greenland, we use samples from the Greenland Ice Sheet Project 2 (GISP2).

We shall refer to late Holocene data from these sites as the calibration dataset, analogous to a core top data set in the sediment coring literature. Site characteristics, coordinates, and number of samples included in the calibration data set are given in Table 1. The DE08-OH site is a recent revisit of the Law Dome DE08 site. The DE08-OH core was measured at sub-annual resolution to understand cm-scale $^{86}Kr_{xs}$ variations due to for example layering in firn density and bubble trapping (Appendix B). In addition to the calibration data set, we present a record of Kr-86 excess going back to the LGM from WDC.

### 2.2 Ice sample analysis

We broadly follow analytical procedures described elsewhere (Bereiter et al., 2018a; Bereiter et al., 2018b; Headly and Severinghaus, 2007; Severinghaus et al., 2003). In short, an 800 g ice sample, its edges trimmed with a band saw to expose fresh surfaces, is placed in a chilled vacuum flask that is then evacuated for 20 minutes using a turbomolecular pump. Air is extracted from the ice by melting the sample while stirring vigorously with a magnetic stir bar, led through a water trap, and cryogenically trapped in a dip tube immersed in liquid He. Next, the sample is split into two unequal fractions. The smaller fraction (about 2% of total air) is analyzed for $\delta^{15}$N-$N_2$, $\delta^{18}$O-$O_2$, $\delta O_2/N_2$ and $\delta Ar/N_2$ on a 3kV Thermo Finnigan Delta V plus dual inlet IRMS (isotope ratio mass spectrometer). In the larger fraction, noble gases are isolated via hot gettering to remove reactive gases. The purified noble gases are then analyzed for $\delta^{40/36}$Ar, $\delta^{40/38}$Ar, $\delta^{86/82}$Kr, $\delta^{86/84}$Kr, $\delta^{86/83}$Kr, $\delta Kr/Ar$ and $\delta Xe/Ar$ on a 10kV Thermo Finnigan MAT253 dual inlet IRMS. We reject one sample from RICE due to incomplete sample transfer, and one sample from WDC due to problems with the water trap. Calibration is done for each measurement campaign by running samples of La Jolla pier air.

All calibration (core top) data were measured using "Method 2" as described by Bereiter et al. (2018a), with a longer equilibration time during the splitting step than used in that study to improve isotopic equilibration between the fractions. The exception is the DE08-OH site, where the ice sample (rather than the extracted gas sample) was split into two fractions – the advantage of this approach is that it does not require a gas splitting step that is time-consuming and may fractionate the isotopes; the downside is that the samples may have slightly different isotopic composition due to the stochastic nature of bubble trapping and the different gas-loss histories of the ice pieces.

Measurements of the WDC downcore data set were performed over five separate measurement campaigns that occurred in February-April 2014, February-April 2015, August 2015, August 2020, and August 2021,



respectively. The first three campaigns are described by Bereiter et al. (2018b), in which the $^{86}Kr_{xs}$ data are
a by-product of measuring $\delta Kr/N_2$ for reconstructing global mean ocean temperature. Campaigns 1 and 2
are in good agreement, whereas campaign 3 appears offset from the other two by an amount that exceeds
the analytical precision (offset around 35 per meg ‰$^{-1}$). To validate the main features in the record, we
performed two additional campaigns (4 and 5), in which all the gas extracted from each ice sample was
quantitatively gettered and only analysed for Ar and Kr isotopic composition. The downcore record, as well
as the five analytical campaigns, are discussed in detail in section 5.1. Data from the bubble-clathrate
transition zone (here 1000 to 1500 m depth, or ~4ka to 7ka BP are excluded owing to the potential for
artefacts.
All samples were analyzed at Scripps Institution of Oceanography, USA, with the exception of the EDC
samples which were analyzed at University of Bern, Switzerland (Baggenstos et al., 2019). Some of the
EDC samples analyzed had clear evidence of drill liquid contamination, which acts to artefactually lower
$^{86}Kr_{xs}$; the late Holocene data used here were not flagged for drill liquid contamination.
The $2\sigma$ analytical precision of the $\delta^{15}N$, $\delta^{40}Ar$, and $\delta^{86}Kr$ measurements is around 3, 5 and 26 per meg,
respectively, based on the reproducibility of La Jolla Air measurements. Via standard error propagation,
this results in a ~ 22 per meg ‰$^{-1}$ ($2\sigma$) analytical uncertainty for both $^{86}Kr_{xs40}$ and $^{86}Kr_{xs15}$. We have no true
(same-depth) replicates to assess the reproducibility of $^{86}Kr_{xs}$ measurements experimentally. The measured
isotope ratios are corrected for gas loss ($\Delta^{40}_{GL}$) and thermal fractionation ($\Delta^{86}_{TF}$, $\Delta^{40}_{TF}$, $\Delta^{15}_{TF}$) before
interpretation; details on these corrections are given in appendix A. For the coretop calibration study, the
average magnitude of the gas loss and thermal fractionation corrections is +14 and -15 per meg ‰$^{-1}$ in
$^{86}Kr_{xs}$, respectively.
Our study includes two ice cores from the Antarctic Peninsula: BP (2 ice samples) and JRI (5 ice samples).
Measured $\delta Xe/N_2$ ratios (and to a lesser extent the $\delta Kr/N_2$ ratios) in all samples from both locations are
significantly elevated above the expected gravitational enrichment signal (Fig. A1A), which is clear
evidence for the presence of refrozen meltwater in these samples (Orsi et al., 2015). Like xenon, krypton is
highly soluble in (melt)water, and therefore $^{86}Kr_{xs}$ cannot be reliably measured in these samples; we reject
all samples from the BP and JRI sites. It is notable that all samples from both sites show evidence of refrozen
meltwater, given that the high-accumulation BP core is nearly entirely free of visible melt layers, and that
we carefully selected samples without visible melt features at JRI. Visible ice lenses form only when
meltwater pools and refreezes on top of low-permeability layers such as wind crusts; our observations
suggest meltwater can also refreeze throughout the firn in a way that cannot be detected visually.



## 3 Calibrating Kr-86 excess

The $^{86}Kr_{xs}$ proxy for synoptic activity was first proposed on theoretical grounds by Buizert and Severinghaus (2016) – here we provide the first experimental validation of this proxy using a coretop calibration of $^{86}Kr_{xs}$ using late-Holocene ice core samples from nine locations around Antarctica and one in Greenland that represent a wide range of pressure variability in the modern climate.

### 3.1 Spatial variation in synoptic-scale pressure variability

Kr-86 excess is sensitive to air movement (both upward and downward), which in turn is controlled by the magnitude of relative air pressure change. Let $p_i$ be a time series of (synoptic-scale) site surface pressure with $N$ data points, time resolution $\Delta t$, and mean value $\bar{p}$. The time series can span a month, year, or multi-year period, with $\bar{p}$ potentially different for each month or year. We define the parameter $\Phi$ as:

$$\Phi = \frac{1}{N\bar{p}} \sum_{i=1}^{N} \left| \frac{p_i - p_{i-1}}{\Delta t} \right| \tag{5}$$

which we here express in convenient units of % day$^{-1}$. $\Phi$ reflects the intensity of barometric pumping in the firn column. Note that $\Delta t$ should be larger than ~1 hour (the timescale for the entire firn column to equilibrate with the surface pressure), and smaller than about a day (in order to adequately resolve synoptic-scale pressure events). Here we use ERA-interim reanalysis data from 1979-2017 with $\Delta t$ = 6 hours (Dee et al., 2011), from which we calculate monthly and annual $\Phi$ values using Eq. (5). A map of annual-mean $\Phi$ across Antarctica is given in Fig. 2A. At all sites considered, $\Phi$ has a strong seasonal cycle with pressure variability/storminess being strongest in the local winter season (Fig. 2C). Interannual variability in $\Phi$ is greatest along the Siple coast and coastal West Antarctica (Fig. 2B), mainly reflecting the influence of central Pacific (ENSO, PDO) climate variability (Section 4).

### 3.2 Kr-86 excess proxy calibration

Present-day Antarctica has a wide range of $\Phi$ (Fig. 2A), which allows us to validate and calibrate $^{86}Kr_{xs}$. In Fig. 3A we plot the site mean $^{86}Kr_{xs40}$ (with ±1σ error bars) as a function of $\Phi$. We find a Pearson correlation coefficient of $r$ = -0.94 when using site mean $^{86}Kr_{xs40}$, and $r$ = -0.83 when using the $^{86}Kr_{xs40}$ of individual samples, respectively ($p$ < 0.01). Note that in this particular case the site-mean $^{86}Kr_{xs40}$ and $^{86}Kr_{xs15}$ are identical (because by design, after thermal correction $\delta^{15}N = \delta^{40}Ar$); the error bars are different, though.

The $^{86}Kr_{xs}$ data have been corrected for gas loss (Appendix A1) and thermal fractionation (Appendix A2); with the gas loss correction being the more uncertain component. Figure 3B shows the correlations of the calibration curve as a function of the gas loss scaling parameter $\varepsilon_{40}$. We find a good correlation over a wide range of $\varepsilon_{40}$ values, proving our calibration is not dependent on the choice of $\varepsilon_{40}$. When using uncorrected $^{86}Kr_{xs40}$ data the site mean correlation is $r$ = -0.71; when applied individually, both the gas loss and thermal correction each improve the correlation to $r$ = -0.77 and $r$ = -0.79, respectively (Fig. A3, all $p$ < 0.05). Based on these tests we conclude that the observed relationship is not an artefact of the applied corrections. The applied corrections improve the correlation, which increases confidence in the method. The calibration results for $^{86}Kr_{xs15}$ are shown in Fig. A4.



Notably, there is a large spread in $^{86}Kr_{xs}$ across samples from a single site, particularly at the high $\Phi$ sites
of SDM and RICE (note the $\pm1\sigma$ error bars). This spread is larger than the measurement uncertainty, and
we believe this variance reflects a signal that is truly present in the ice. The Siple coast and Roosevelt Island
experience the largest $\Phi$ interannual variability in Antarctica (Fig. 2B), and it is therefore likely that our
coarse sampling is aliasing the true $^{86}Kr_{xs}$ signal. The variance in $^{86}Kr_{xs}$ may contain climate information
also; this is reminiscent of the way in which the variance (rather than the mean) of $\delta^{18}O$ in individual
planktic foraminifera in ocean sediment samples from the equatorial Pacific can been used as a proxy for
past ENSO variability (Koutavas et al., 2006).
Both theoretical considerations and observations thus suggest $^{86}Kr_{xs}$ is a proxy for barometric surface
pressure variability at the site, and in the remainder of this manuscript we will interpret it as such.

### 3.3 Discussion of the Kr-86 excess proxy

Our interpretation of $^{86}Kr_{xs}$ as a proxy for pressure variability is somewhat complicated by the possibility
of deep convective zones, which have the same $^{86}Kr_{xs}$ signature as barometric pumping. This was
discovered at the Megadunes (MD) site, central East Antarctica; at this zero-accumulation site deep cracks
form in the firn layer that facilitate a 23 m deep convection zone (Severinghaus et al., 2010). In fact, this
observation led earlier work to suggest that noble gas gravitational disequilibrium may be used as a proxy
for convective zone thickness (Kawamura et al., 2013), rather than synoptic-scale pressure variability as
suggested here. Although megadunes and zero-accumulation zones are ubiquitous and cover 20% of the
Antarctic Plateau (Fahnestock et al., 2000), ice cores are seldom drilled in these areas and it is safe to
assume that they never formed at sites like WAIS Divide that had relatively high accumulation rates even
during the last glacial period. Performing the corrections for thermal and size-dependent fractionation is
challenging at MD, and we suggest that the MD $^{86}Kr_{xs}$ is in the range of -2 to -55 per meg ‰$^{-1}$; even at the
larger limit, this is still smaller in magnitude than $^{86}Kr_{xs}$ anomalies at several modern-day sites with small
convective zones (such as SDM, RICE and the Law Dome sites), suggesting barometric pumping is capable
of producing larger $^{86}Kr_{xs}$ signals than even the most extreme observed case of convective surface mixing.
Having $^{86}Kr_{xs}$ measured in MD ice core (rather than firn air) samples would be valuable for a more
meaningful comparison to the ice core sample measurements presented here. Windy sites can have
substantial convective zones of ~ 14 m (Kawamura et al., 2006), and future studies of $^{86}Kr_{xs}$ at such sites
would be valuable.
Currently, 1-D and 2-D firn air transport model simulations underestimate the magnitude of the $^{86}Kr_{xs}$ signal
compared to measurements in mature ice samples (Birner et al., 2018), complicating scientific
understanding of the proxy. In these models, the effective molecular diffusivity of each gas is scaled linearly
to its free air diffusivity. The ratio of krypton to argon free air diffusivity is 0.78. This ratio, which directly
sets the magnitude of the simulated $^{86}Kr_{xs}$, may actually be smaller than 0.78 in real firn, as krypton is more
readily adsorbed onto firn surfaces retarding its movement (similar to gasses moving through a gas
chromatography column). This may be one explanation for why models simulate too little $^{86}Kr_{xs}$.
Another likely explanation for the model-data mismatch is that certain critical sub-grid processes (such as
the aforementioned pore-size dependence of the Péclet number) are not adequately represented in these
models. Barometric pumping may further actively shape the pore network through the movement of water
vapor, thereby keeping certain preferred pathways connected and open below the density where percolation





theory would predict their closure (Schaller et al., 2017). The fate of a pore restriction is determined by the
balance between the hydrostatic pressure (that acts to close it) and vapor movement away from its convex
surfaces (that acts to keep it open); we speculate that barometric Darcy air flow keeps high-flow channels
connected longer by eroding convex surfaces. This enhances the complexity (and therefore dispersivity) of
the deep firn pore network and possibly creates a non-linear $^{86}Kr_{xs}$ response to barometric pumping. The
hypothesized channel formation in deep firn is driven by a positive feedback on flow volume, and somewhat
reminiscent of erosion-driven stream network formation in fluvial geomorphology.
Firn models predict that the gravitational disequilibrium effect in elemental ratios (such as $\delta Kr/Ar$) should
be proportional to that in isotopic ratios. However, the observations suggest that the former is usually
smaller than would be expected from the latter. We do not have an explanation for this effect. Including
measurements of xenon isotopes and elemental ratios in future measurement campaigns may be able to
provide additional constraints to better understand this discrepancy.
Measurements on firn air samples, where available, suggest a smaller $^{86}Kr_{xs}$ anomaly in firn air than found
in ice core samples from the same site. We attribute this in part to a seasonal bias that is introduced by the
fact that firn air sampling always takes place during the summer months, whereas the synoptic variability
that drives the Kr-86 excess anomalies is largest during the winter (Fig. 2C); consequently, firn air
observations are biased towards weaker $^{86}Kr_{xs}$. Further, in the deep firn where $^{86}Kr_{xs}$ anomalies are largest,
firn air pumping may not yield a representative air sample, but rather be biased towards the well-connected
porosity at the expense of poorly-connected cul-de-sac-like pore clusters. Since barometric pumping
ventilates this well-connected porespace with low-$^{86}Kr_{xs}$ air from shallower depths, the firn air sampling
may not capture a representative $^{86}Kr_{xs}$ value of the full firn air content. These explanations are all somewhat
speculative, and a definitive understanding of the firn-ice differences is lacking at this stage.
Gas loss and thermal corrections are critical to the interpretation of $^{86}Kr_{xs}$. The thermal correction is applied
to account for thermal gradients in the firn ($\Delta T$, here defined as the temperature at the top minus the
temperature at the base of the firn), which are chiefly caused by geothermal heat or surface temperature
changes at the site. At low-accumulation sites geothermal heating leads to $\Delta T < 0$. We use $^{15}N$-excess ($\delta^{15}N$
$- \delta^{40}Ar/4$) to estimate the thermal gradient in the firn (Appendix A2). Because nitrogen and argon have
similar diffusivities but different thermal diffusion coefficients, $\delta^{15}N$ - $\delta^{40}Ar$ is relatively insensitive to
barometric pumping yet sensitive to thermal fractionation, allowing estimating $\Delta T$.
Besides the actual thermal gradients in the firn, the isotopic composition may also be impacted by seasonal
rectifier effects. If the firn air transport properties differ between the seasons (for example due to thermal
contraction cracks, convective instabilities, or seasonality in wind pumping), this can result in a thermal
fractionation of isotopic ratios in the absence of a thermal gradient (Morgan et al., 2022).
For the WDC, DSS and GISP2 sites we obtain $\Delta T$ values close to zero as expected for these high-
accumulation sites; for the SP, SDM, RICE, and DF sites we find $\Delta T$ ranging from -0.76 to -1.18°C, in
agreement with the effect of geothermal heat. The high-accumulation DE08 and DE08-OH sites both have
an unexpectedly large $\Delta T$ of -1.6°C; the good agreement between the sites suggest it is likely a real signal,
yet we can rule out geothermal heat as the cause. This may suggest that the Law Dome DE08 site is subject
to a seasonal rectifier effect, or a recent climatic cooling. Last, the EDC site shows an unexpected $\Delta T =$



$+1.6 \pm 1.89^{\circ}C$. Three possible explanations are: (1) the aforementioned drill liquid contamination for this
core; (2) a summertime-biased seasonal rectifier; or (3) an over-correction of $\delta^{40}Ar$ for gas loss, which
could occur for example if natural and post-coring fugitive gas loss fractionate $\delta^{40}Ar$ differently and EDC
samples were impacted mostly by the former type (our correction is mostly based on measurements of the
latter type).
For the Law Dome DE08-OH site we observe large (5-fold) sub-annual variations in $^{86}Kr_{xs}$ (Fig. B1). The
magnitude of the $^{86}Kr_{xs}$ layering is truly remarkable. The isotopic enrichment of each gas ($\delta^{15}N$, $\delta^{40}Ar$,
$\delta^{86}Kr$) can be converted to an effective diffusive column height (DCH). For the samples with the smallest
(greatest) $^{86}Kr_{xs}$ magnitude, this DCH is around 1 m (6 m) shorter for $\delta^{86}Kr$ than it is for $\delta^{15}N$. The firn air
transport physics that may explain such phenomena are beyond our current scientific understanding. The
sub-annual variations may be related to the seasonal cycle in storminess (Fig. 2C), though that seems
improbable as the gas age distribution at the depth of bubble closure has a width of several years (Schwander
et al., 1993). Another reason may be seasonal layering in firn properties – such as density, grain size, and
pore connectivity – that control the degree of disorder and dispersive mixing occurring in the firn, and lead
to a staggered firn trapping and seasonal variations in Δage (Etheridge et al., 1992; Rhodes et al., 2016).
The sample air content estimated from the IRMS inlet pressure is similar for all measurements, making it
unlikely that the variations in $^{86}Kr_{xs}$ are caused by remnant open porosity in lower-density layers. In any
case it is remarkable that such large variations in gas composition can arise and persist on such small length
scales, given the relatively large diffusive, dispersive, and advective transport length scales of the system.
More work is needed to establish the origin of the sub-annual variations in ice core $^{86}Kr_{xs}$.
Another puzzling observation is the positive $^{86}Kr_{xs}$ at the Dome Fuji (DF) site; theoretical considerations
suggest it should always be negative. In part this may be due to an over-correction of $\delta^{40}Ar$ for gas loss,
which would act to bias $^{86}Kr_{xs}$ in the positive direction. This correction is largest at DF owing to the very
negative $\delta O_2/N_2$ and $\delta Ar/N_2$ (Fig. A1); while we base our correction on published work, it is conceivable
that we overestimate the true correction (Appendix A1). In particular, our gas loss correction is based on
observations on artefactual post-coring gas loss, which may fractionate $\delta^{40}Ar$ differently than natural
fugitive gas loss during bubble close-off. Omitting the gas loss correction indeed makes $^{86}Kr_{xs}$ at DF
negative (Fig. A3C-D). Another hypothesis is that the positive $^{86}Kr_{xs}$ signal is an artefact of the seasonal
rectifier that Morgan et al. (2022) identify at DF. In this work we assume a linear approach in which the
effect of the rectifier can be described by a single $\Delta T$ value that is the same for isotopic pairs. In reality,
there may be non-linear interactions between thermal fractionation and firn advection that impact the
isotopic values of the various gases in a more complex way than captured in our approach.
The $^{86}Kr_{xs}$ is also correlated with other site characteristics besides $\Phi$. For site elevation we find $r = 0.96$
(0.84); and for mean annual temperature $r = -0.87$ (-0.76); the number in parentheses gives the correlation
when using all the individual samples rather than site-mean $^{86}Kr_{xs}$. The listed correlations all have $p < 0.01$.
For site accumulation we do not find a statistically significant correlation at the 90% confidence level. The
correlations with elevation and temperature are comparable to those we find for $\Phi$; this is no surprise given
that elevation, $\Phi$ and $T$ are all strongly correlated with one another, mainly because elevation directly
controls both $T$ (via the lapse rate) and $\Phi$ (by limiting the penetration of storms). To our knowledge there
are no mechanisms through which either elevation or annual-mean temperature could drive kinetic isotopic
fractionation in the firn layer. Perhaps other unexamined site characteristics (such as the degree of density





layering, or the magnitude of the annual temperature cycle) could provide good correlations also, suggesting additional hidden controls on $^{86}Kr_{xs}$. The data needed to assess such hidden controls are not available for most sites.

The calibration of the $^{86}Kr_{xs}$ proxy is based on spatial regression. In applying the proxy relationship to temporal records, we make the implicit assumption that proxy behavior in the temporal and spatial dimensions is at least qualitatively similar. This assumption may prove incorrect. In particular, changes in insolation are known to impact firn microstructure and bubble close-off characteristics, which in turn impacts gas records of $\delta O_2/N_2$ and total air content (Bender, 2002; Raynaud et al., 2007). Since $^{86}Kr_{xs}$ is linked to the dispersivity of deep firn, it seems probable that insolation has a direct impact on $^{86}Kr_{xs}$ also via the firn microstructure. We will revisit this issue in our interpretation of the WD $^{86}Kr_{xs}$ record (Section 5). Overall, we anticipate $^{86}Kr_{xs}$ to be a qualitative proxy for synoptic variability, yet want to caution against quantitative interpretation based on the spatial regression slope.

The observations presented in this section clearly highlight the fundamental shortcomings of our current understanding of firn air transport hinting at the existence of complex interactions, presumably at the pore-scale, that are not being represented. Percolation theory finds that near the critical point (presumably the lock-in depth) a network becomes fractal in its nature; we suggest that this fractal nature of the pore network likely contributes to non-linear pore-scale interactions that give rise to the $^{86}Kr_{xs}$ observations in ice. While the observed correlation of Fig. 3C is highly encouraging, further work is critical to understand this proxy. Examples of such future studies are: (1) additional high-resolution records that can resolve the true variations that exist in a single ice core, similar to the DE08-OH record; (2) 3-D firn air transport model studies; (3) improvements to the gas loss correction; (4) additional coring sites to further confirm the validity of the proxy; (5) Adding xenon isotopic constraints ($^{136}Xe$ excess) as an additional marker of isotopic disequilibrium; (6) numerical simulations of pore-scale air transport in large-scale firn networks; (7) experimental studies of dispersion in firn samples; and (8) percolation theory approaches to study the fractal nature of the pore network of the lock-in zone.



## 4 Present-day controls on Kr-86 excess in Antarctica

In this section we investigate the large-scale patterns of climate variability in the Southern Hemisphere that could affect $\Phi$ and therefore $^{86}Kr_{xs}$ over Antarctica. We begin by investigating the patterns in the wind field that are associated with changes in $\Phi$ at ice core sites, before examining how more canonical patterns of Southern Hemisphere climate variability, such as the southern annular mode (SAM), might affect $\Phi$ over the whole of Antarctica.

We use ERA-interim reanalysis data for the 1979-2017 period (Dee et al., 2011) to evaluate the present-day controls on synoptic-scale pressure variability in Antarctica. Kr-86 excess in an ice core sample averages over several years of pressure variability, and therefore we focus on annual-mean correlation in our analysis. The annual-mean $\Phi$ is calculated from the 6-hourly reanalysis data using Eq. (5). Note that we let the year run from April to March to avoid dividing single El Niño / La Niña events across multiple years.

At all Antarctic sites investigated, a similar pattern exists; four representative locations are shown in Fig. 4, where we regress the zonal wind in the lower (850 hPa, color shading) and upper troposphere (200 hPa, contours) onto our surface pressure variability parameter $\Phi$. We find that synoptic pressure variability at these sites is linked to zonal winds along the southern margin of the eddy-driven subpolar jet (SPJ), which extends from the surface to the upper troposphere (Nakamura and Shimpo, 2004; Trenberth, 1991). Sites near the ice sheet margin (Figs. 4A, B and D) are most sensitive to the SPJ edge in their sector of Antarctica, whereas interior sites (Fig. 4C) appear sensitive to the overall strength/position of the SPJ. Note that strengthening, broadening or southward shifting of the SPJ all can in principle enhance site $\Phi$.

Pressure variability at WDC is furthermore correlated with the strength of the Pacific Subtropical jet (STJ) aloft (solid contour lines centered around 30ºS in the Pacific in panel 4A), forming an upper troposphere wind pattern that resembles the wintertime South Pacific split jet (Bals-Elsholz et al., 2001; Nakamura and Shimpo, 2004); this agrees with the finding that a strengthening of the split jet enhances storminess over West Antarctica (Chiang et al., 2014).

Next, we investigate how the well-known patterns of large-scale atmospheric variability, such as SAM and ENSO, impact pressure variability in Antarctica. Figure 5 shows the correlation of $\Phi$ with the three leading modes of SH extra-tropical atmospheric variability; the correlation with various indices and modes for individual ice core locations is given in Table 2. Most teleconnection patterns have a specific season during which they are strongest; here we do not differentiate between seasons, because $^{86}Kr_{xs}$ in ice core samples averages over all seasons.

Globally, annual-mean $\Phi$ is highest over the Southern Ocean (Fig. 5A); a region of enhanced baroclinicity associated with the eddy-driven SPJ (Nakamura and Shimpo, 2004). The green line denotes the latitude of maximum $\Phi$, corresponding roughly to the latitude with the highest storm track density (57.8ºS on average).

The dominant mode of atmospheric variability in the SH extratropics is the southern annular mode, representing the vacillation of atmospheric mass between the mid- and high-latitudes (Thompson and Wallace, 2000). Figure 5B shows 500 hPa geopotential height (Z500) anomalies associated with the SAM as contours, with the color shading giving the correlation between $\Phi$ and the SAM index. During the



positive SAM phase (negative Z500 over Antarctica) we find that the stormtracks and maximum synoptic activity are displaced towards Antarctica (positive $\Phi$ correlation poleward of the green line in Fig. 5B). This is associated with a strengthening and poleward displacement of the SH westerly winds that occurs during a positive SAM phase. More locally, $\Phi$ on the Antarctic Peninsula is positively correlated with the SAM-index (Table 2); $\Phi$ at the other sites is not meaningfully impacted. This suggests that the variations associated with the SAM (as commonly defined) do not extend far enough poleward to meaningfully impact Antarctica with the exception of the Peninsula. Enhanced synoptic variability on the Peninsula during positive SAM phases is consistent with observations of enhanced snowfall at those times (Thomas et al., 2008).

The second mode of SH extratropical variability is the Pacific-South American Mode 1 (PSA1), which reflects a Rossby wave response to sea surface temperature (SST) anomalies over the central and eastern equatorial Pacific (Mo and Paegle, 2001), and is therefore closely linked to ENSO on interannual time scales (we find a correlation of $r = 0.77$ between the annual mean PSA1 and Niño 3.4 indices). $\Phi$ in the Amundsen and Ross Sea sectors (WDC, SDM and RICE) is positively correlated to the PSA1 and Niño 3.4 SST, suggesting larger synoptic activity during El Niño phases and low activity during La Niña phases. The PSA2 pattern, also linked to SST anomalies in the tropical Pacific (Mo and Paegle, 2001), is likewise correlated to $\Phi$ in the Amundsen and Ross Sea sectors (Fig. 5C and Table 2). While all the correlations listed are statistically significant, they explain only a fraction of the total variability.

Next, we consider anomalies in sea ice area and extent (Parkinson and Cavalieri, 2012). We focus on the Ross and Amundsen-Bellingshausen Seas where impacts on WAIS Divide may be expected. At the 90% confidence level we do not find significant correlations to sea ice area or extent at most core locations (Table 2). Correlations to sea ice extent are (even) weaker than those for sea ice area and consequently not shown. We performed a lead-lag study of the correlations between $\Phi$ and sea ice area/extent in the various sectors, and find that in all cases maximum correlations occur for the sea ice changes lagging 0 to 4 months behind $\Phi$; we interpret this to mean that the sea ice is responding to changes in atmospheric circulation, rather than driving them.

Overall, we find that synoptic activity at WAIS Divide, the site of most interest here, is controlled by the position and/or strength of the stormtracks at the southern edge of the SPJ in the Pacific sector of the Southern Ocean (Ross, Amundsen and Bellingshausen Seas), with little sensitivity to the SPJ behavior in the other sectors. Owing to its remote southern location, WDC is only weakly impacted by the commonly-defined large-scale modes of atmospheric variability. Most notably, WDC has a modest influence from the tropical Pacific climate, as shown by a correlation around $r \approx 0.3$ to the PSA1, Niño 3.4 and PDO indices (Table 2). We further find statistically significant correlations (up to $r = 0.44$) between WDC $\Phi$ and SST in broad regions of the central and eastern tropical Pacific (not shown). We suggest that ENSO weakly impacts storminess at WDC (around 10% of variance explained) via its impact on the SPJ in the South Pacific.





**5 Barometric variability in West Antarctica during the last deglaciation**

**5.1 The 0-24 ka WAIS Divide Kr-86 excess record**

The WAIS Divide downcore $^{86}Kr_{xs}$ dataset we present here was produced during five separate measurement campaigns that occurred in February-April 2014, February-April 2015, August 2015, August 2020, and August 2021, respectively. Campaigns 1-3 were reported previously (Bereiter et al., 2018a; Bereiter et al., 2018b), and campaigns 4 and 5 were meant to resolve conflicts between the $^{86}Kr_{xs}$ data sets from these earlier campaigns. Three slightly different measurement approaches were used. Campaign 1 uses "Method 1" from Bereiter et al. (2018a), in which the air sample splitting is done in a water bath for over 12 hours to equilibrate the sample. Campaigns 2 and 3 use "Method 2" from Bereiter et al. (2018a), in which a bellows is used to split the air samples for over 4 to 6 hours. Campaigns 4 and 5 do not involve splitting of the air sample, and only analyzed the Kr and Ar isotopic ratios. During campaign 4 a glass bead from the water trap had gotten stuck in the tubing, restricting the flow and likely resulting in incomplete air extraction from the melt water.

Figure 6 compares $^{86}Kr_{xs40}$ (panel A) and $^{86}Kr_{xs15}$ (panel B) from the five campaigns. Campaign 1 is the only campaign that spans the full age range of the record, making it the most valuable of the three campaigns. Campaigns 2 and 3 are mostly restricted to the Pleistocene and Holocene periods respectively, with little overlap between them. Campaigns 4 and 5 aimed to reproduce some of the most salient features in the earlier three.

No true replicate samples were analyzed between the campaigns, in part because the large sample size requirement precludes this. To assess offsets, we rely on nearest-neighbor linear interpolation. We find an offset of 5 per meg between the first and second campaign (during their period of overlap); this is within the analytical precision (22 per meg), suggesting these two campaigns are in good agreement. The agreement is good for both the $^{86}Kr_{xs40}$ and $^{86}Kr_{xs15}$ definitions. The first downcore campaign furthermore overlaps in depth with the WDC calibration dataset (gray data in Fig. 6); we find no offset between those data sets either. Data from campaign 2 appear to have more scatter, possibly reflecting the shorter equilibration time during sample splitting.

We combine data from the first two campaigns, and evaluate their offset to data from the other three campaigns using nearest-neighbor linear interpolation. For campaigns 3, 4 and 5 we find an offset of -31, -22 and -23 per meg ‰$^{-1}$ in $^{86}Kr_{xs40}$, respectively. For campaign 3 the offset is -34 per meg ‰$^{-1}$ in $^{86}Kr_{xs15}$. It is remarkable that all three later campaigns are more negative in $^{86}Kr_{xs}$ than the first two. Campaign 3 shows the greatest offset (greater than analytical precision), and has more scatter in both $^{86}Kr_{xs}$ (Fig. 6) and $^{15}N$ excess (Fig. A5), and less care was taken during this campaign that the IRMS conditions were stable. The offset of campaign 4 may be attributed to the incomplete sample transfer due to the bead stuck in the line. The offset in campaign 5 is hard to explain. The systematically more negative $^{86}Kr_{xs}$ of campaigns 4 and 5 may reflect sample storage effects, as these were measured 5-6 years after campaign 1 and 2. However this would not explain the negative values of campaign 3. The good $^{86}Kr_{xs}$ agreement between DE08 and DE08-OH, drilled 32 years apart, would also argue against large storage effects. For campaign 4 and 5 only Ar and Kr isotope ratios were measured, and so we lack typical tracers of gas loss ($\delta O_2/N_2$ and $\delta Ar/N_2$.)





In the remainder of this paper we will interpret the combined data from campaigns 1 and 2, but with the
caveat that there is a persistent offset with later campaigns. However, the features we interpret are
corroborated by the later campaigns, if one takes the offset into account. To aid interpretation of the data,
we apply a Gaussian smoothing spline with a smoothing filter width that varies depending on the data
density (from 250-year width in the deglaciation itself where the data density is high, to 1750 years in the
Holocene and LGM where data density is low). To estimate the uncertainty in the smoothing spline we use
a Monte Carlo approach that considers uncertainty in (1) the gas loss correction, by randomly sampling $\varepsilon_{40}$
in the range of 0 to -0.008; (2) the thermal correction, by randomly scaling the thermal scenario (Fig. A5)
by a factor ranging from 0 to 2; and (3) analytical errors, by adding random errors to individual data points
drawn from a normal distribution with a $2\sigma$ width of 22 per meg. The $\pm 1\sigma$ uncertainty range with mean
value are shown as the gray envelope and center line in Fig. 6. We believe the following observations to be
robust:
• The Holocene shows a trend towards increasingly negative $^{86}Kr_{xs}$, suggesting a gradual increase in
synoptic activity toward the present. Minimum synoptic activity in West Antarctica occurs during the
early Holocene around 10 ka BP; the Monte Carlo study suggests $^{86}Kr_{xs40}$ in the early Holocene (8ka-
10ka BP) is 30.5 $\pm$18 per meg ‰$^{-1}$ ($\pm 2\sigma$) below the late-Holocene value (last 2 ka). Using the slope of
our core-top calibration (Fig. 3), we estimate that early-Holocene WDC synoptic activity $\Phi$ is ~17%
weaker than it is today. This change is comparable to the $2\sigma$ magnitude of interannual variations in
annual mean $\Phi$ at the site today (or about half the peak-to-peak variations thereof). This Holocene trend
is seen in the data from campaigns 1, 3 and 4; campaign 5 does not suggest a trend but has only one
late Holocene data point making it less robust.
• The most pronounced change occurs at the Younger Dryas (YD) - Holocene transition, where $^{86}Kr_{xs}$
becomes more positive (by 30.1 $\pm$16 per meg ‰$^{-1}$, comparing YD and early Holocene) implying a
decrease in synoptic activity. This transition is observed in campaigns 1, 2, 4 and 5 that cover this time
period (the third campaign does not cover it), and represents a ~17% drop in synoptic activity ($\Phi$).
• During the Last Glacial Maximum (LGM), WDC synoptic activity was perhaps slightly weaker than at
present, but not significantly so ($^{86}Kr_{xs40}$ more positive by 11 $\pm$13 per meg ‰$^{-1}$). The West Antarctic
ice sheet elevation was likely higher during the LGM, and a 300 m elevation increase would by itself
increase $^{86}Kr_{xs40}$ by 10 per meg ‰$^{-1}$, all else being equal (Appendix A3). This feature is seen in
campaign 1 and not covered by the other campaigns.
• The deglaciation itself has enhanced synoptic activity, in particular during the two North-Atlantic cold
stages Heinrich Stadial 1 (HS1) and the YD as highlighted with yellow bars in Figs. 6 and 7. Synoptic
activity during these periods is enhanced relative to the adjacent LGM and early Holocene, yet
comparable to today. This feature is seen in campaigns 1 and 2, and in 4 and 5 for the transition into
the Holocene.
Below we will interpret the deglacial WD $^{86}Kr_{xs}$ record in terms of barometric variability. Before doing so
we want to emphasize that firn processes may have been imprinted onto the record also, in particular on
orbital timescales where firn microstructure responds to local (summer) insolation intensity (Bender, 2002).
High summer insolation results in more depleted $\delta O_2/N_2$ and reduced air content, likely via stronger
layering and a delayed pore close-off process (Fujita et al., 2009).



Local summer solstice insolation in Antarctica increases through the Holocene, with the highest values in
the late Holocene. This may impact $^{86}Kr_{xs}$, although it is not a-priori clear what the sign of this relationship
would be. The sense of the Holocene temporal trends is that a more negative $^{86}Kr_{xs}$ coincides with more
negative $\delta O_2/N_2$. Note that this is opposite to the trends seen in the spatial calibration, where sites with the
most negative $\delta O_2/N_2$ (DF, SP, EDC) have the most positive $^{86}Kr_{xs}$. For now, the impact of local insolation
on $^{86}Kr_{xs}$ via firn microstructure remains unknown, which is an important caveat in interpreting the orbital-
scale changes in WD $^{86}Kr_{xs}$. The abrupt $^{86}Kr_{xs}$ increase at the Holocene onset is too abrupt to be caused by
insolation changes, and thus we can interpret that change with more confidence.
**5.2 Barometric variability at WAIS Divide during the last deglaciation**
In the present-day, synoptic-scale pressure variability at WAIS Divide is correlated with zonal wind
strength along the southern margin of the SPJ (Section 4). In our interpretation, a more negative $^{86}Kr_{xs}$
reflects a strengthening or southward shift of the SPJ in the Pacific sector. Here we provide a climatic
interpretation of the deglacial WDC $^{86}Kr_{xs}$ record, and suggest that variations in synoptic variability at WDC
are linked to meridional movement of the ITCZ on millennial and orbital timescales.
The main features of the deglacial WDC $^{86}Kr_{xs}$ record listed in Section 5.1 resemble similar features seen
in records of (sub-) tropical hydrology and monsoon strength, such as the speleothem calcite $\delta^{18}O$ records
from Hulu Cave, China (Fig. 7C) and from Botuvera cave, southern Brazil (Fig. 7D), which are thought to
reflect the intensity of the East Asian and South American summer monsoons, respectively (Cruz et al.,
2005; Wang et al., 2007; Wang et al., 2001). These two monsoon records are anti-correlated, showing
opposing rainfall trends between the NH and SH on both orbital and millennial timescales. This pattern is
commonly attributed to displacement of the mean meridional position of the ITCZ (Chiang and Friedman,
2012; McGee et al., 2014; Schneider et al., 2014), driven by hemispheric temperature differences (Fig. 7B).
On orbital timescales such ITCZ migration has a strong precessional component, moving towards the
hemisphere with more intense summer peak insolation; on millennial timescales the ITCZ responds to
abrupt North-Atlantic climate change associated with the D-O and Heinrich cycles (Broccoli et al., 2006;
Chiang and Bitz, 2005; Wang et al., 2001), which are in turn linked to changes in meridional heat transport
by the Atlantic meridional overturning circulation, or AMOC (Lynch-Stieglitz, 2017; Rahmstorf, 2002).
Changes in mean ITCZ position have a strong influence on the structure and strength of the SH jets. During
periods when the NH is relatively cold (such as D-O stadials or periods with negative orbital precession
index) the ITCZ is displaced southward and the SH Hadley cell is weakened, thereby also weakening the
SH upper-tropospheric subtropical jet (Ceppi et al., 2013; Chiang et al., 2014). The reverse is also true, with
the ITCZ shifted northward during NH warmth, associated with a strengthening of the SH Hadley cell and
STJ. In a range of model simulations (Ceppi et al., 2013; Lee and Kim, 2003; Lee et al., 2011; Pedro et al.,
2018) the weakening of the SH STJ (as during NH cold) is furthermore accompanied by a strengthening
and/or southward shift of the SPJ/eddy-driven jet and SH westerly winds. Recently, ice core observations
have confirmed in-phase shifts in the position of the SHW occur during the D-O cycle in parallel to those
of the ITCZ (Buizert et al., 2018; Markle et al., 2017). Marine records of fluvial sediment runoff off the
Chilean coast suggest precession-phased movement of the South Pacific SPJ, again in parallel to the ITCZ
movement (Lamy et al., 2019).



While data and models thus appear to agree on this first-order zonal-mean circulation response, zonal
asymmetries may lead to divergent outcomes at individual locations, particularly in the Pacific sector of
Antarctica where WDC is located. While the Heinrich (i.e. NH cooling) simulations clearly show the
aforementioned zonal-mean strengthening of the eddy-driven jet (Lee et al., 2011), they also suggest a
weakening of the South Atlantic austral winter split jet (Chiang et al., 2014); in this weakened split jet
configuration the STJ and SPJ are weakened at the expense of a strengthened mid-latitude jet. Essentially
the literature presents us with two opposing hypotheses for the response of the South Pacific SPJ to ITCZ
migration. In the zonal-mean framework, meridional ITCZ migration is accompanied by a parallel shift
(and/or strengthening) of the SH SPJ/eddy-driven jet, suggesting an anti-correlation between ITCZ latitude
and Antarctic storminess (with weak synoptic activity as the ITCZ is shifted north). However, if zonal
asymmetries in the SPJ response are considered, storminess at WDC may actually have the opposite
relationship to ITCZ position, due to a proposed weakening (strengthening) of the split jet as the ITCZ
shifts south (north). Our $^{86}Kr_{xs}$ record implies that synoptic activity at WDC is anticorrelated with ITCZ
position, suggesting that the zonally symmetric SPJ response advocated by e.g. Ceppi et al. (2013)
dominates over the zonally asymmetric split jet response advocated by Chiang et al. (2014).
The present-day SAM is sometimes suggested as an analogue for past shifts in the meridional position of
the SHW and eddy-driven jet (Rind et al., 2001); during positive SAM phases the SHW are displaced
poleward, and during negative phases equatorward. The WDC Kr-86 excess record, combined with our
analysis of the present-day circulation (Fig. 4), implies changes to the position and/or strength of the
southern edge of the SPJ. However, we find that the present-day SAM does not have a statistically
significant impact on synoptic variability at WDC (Table 2). Perhaps the SAM is not a good analogue for
these past changes in circulation after all, in particular when considering the impact of SHW shifts on
Antarctic storminess. The present-day SAM represents a mode of internal variability, with anomalies
persisting for only weeks to months – the timescale is longest in late spring and early summer reflecting a
stronger planetary wave–mean flow interaction (Simpson et al., 2011; Thompson and Wallace, 2000). By
contrast, the shifts in the ITCZ, and presumably the associated changes to the SH jet structure, persist for
centuries to millennia. Moreover, the atmospheric dynamics of the SAM and the ITCZ-driven shifts in the
SHW are very different, with the latter being driven from the tropics via hemispherically asymmetric
changes in Hadley cell and STJ strength. At first glance it may appear contradictory to state, as we do, that
synoptic activity at WDC is not sensitive to the SAM while also suggesting that during the last deglaciation
synoptic activity at WDC is linked to changes in the position of the SH eddy-driven jet and westerlies.
Based on the considerations above, both claims may be true without contradiction.
Besides secular changes to the SPJ position/strength linked to meridional ITCZ movement, WDC $^{86}Kr_{xs}$
may also have imprints from ENSO and tropical Pacific climate. Our analysis suggests a weak, but
statistically significant link to common ENSO indicators (Table 2). Increased synoptic activity at WDC is
linked to enhanced convection in the central and eastern tropical Pacific, which may be due to enhanced
frequency or intensity of El Niño events, or a mean climate state that is more El Niño-like; it seems likely
that the Pacific mean state and ENSO variability are strongly linked (Salau et al., 2012), and the distinction
may be irrelevant.
The key features of the WDC $^{86}Kr_{xs}$ record are compatible with paleo-ENSO changes commonly described
in the literature. A majority of Holocene ENSO reconstructions (Conroy et al., 2008; Driscoll et al., 2014;
Koutavas et al., 2006; Moy et al., 2002; Riedinger et al., 2002; Sadekov et al., 2013) and a wide range of



climate model simulations (Braconnot et al., 2012; Cane, 2005; Clement et al., 2000; Liu et al., 2000; Liu
et al., 2014; Zheng et al., 2008) all suggest weakened ENSO activity during the early and mid-Holocene, a
time with reduced WDC synoptic activity. For example, Fig. 7F shows the number of El Niño events per
century (with trend line) reconstructed from inorganic clastic laminae in sediments from Laguna
Pallcacocha, Ecuador, a region strongly affected by ENSO (Moy et al., 2002). Likewise, it has been
suggested that the SST gradient between the West Pacific warm pool and East Pacific cold tongue was
enhanced during the mid-Holocene, perhaps indicating a more La Niña-like mean climate state (Koutavas
et al., 2002; Sadekov et al., 2013).
Going from the early Holocene to the Younger Dryas (YD), we observe a large increase in WDC synoptic
activity. Enhanced ENSO activity during Heinrich stadials is generally supported by climate model
simulations (Braconnot et al., 2012; Merkel et al., 2010; Timmermann et al., 2007), and by limited proxy
evidence for stadial periods more broadly (Stott et al., 2002). Enhanced ENSO variability during the
deglaciation is also found by Sadekov et al. (2013), although their record lacks the temporal resolution to
resolve the individual stages. The zonal SST gradient in the equatorial Pacific further reaches a minimum
during HS1, also consistent with higher El Niño intensity (Sadekov et al., 2013).
The observed variations in $^{86}Kr_{xs}$ and implied changes in WDC synoptic activity may thus have two
contributions: (1) ITCZ-driven changes to the South Pacific SPJ position, and (2) changes to ENSO activity.
Based on previous work, we argue these two amplify one another in driving WDC storminess, yet we expect
the former to make the larger contribution. To disentangle zonally-uniform changes to the SPJ from changes
specific to the Pacific sector (such as ENSO and the split jet), $^{86}Kr_{xs}$ records from different sectors of
Antarctica are needed. Replication of the deglacial and Holocene WDC $^{86}Kr_{xs}$ record presented here is also
a high priority, both at WDC itself and at the nearby SDM and RICE cores, to validate that the signals we
describe and interpret here are indeed real and regional in scale.
The position of the SHW during the LGM has been a topic of much scientific inquiry. Proxy data have been
interpreted to show a northward LGM shift of the SHW – with other scenarios, including no change at all,
not excluded by the data (Kohfeld et al., 2013). Such a shift is not supported by most climate models (Rojas
et al., 2009; Sime et al., 2013). Our $^{86}Kr_{xs}$ record suggests LGM synoptic activity in West Antarctica to be
comparable to today after accounting for site elevation effects. This would be consistent with a Pacific SPJ
position similar to today. Note that our site is mostly sensitive to the position of the southern edge of the
SPJ, and cannot meaningfully constrain changes to the seasonality, width, and/or northern edge of the
stormtracks. Therefore, it is not a-priori clear whether our observations can be extrapolated to more general
statements about SHW position and strength during the LGM. Our data suggest that SPJ movement follows
insolation and the ITCZ position, and hence the LGM period may not be a good target for studying SHW
movement in the first place given that it has a precession index similar to the present-day.
Changes to the SPJ and its associated westerly surface winds have implications for ocean circulation and
marine productivity in the Southern Ocean via wind-driven upwelling. Opal flux records from the Antarctic
zone (Fig. 7G), reflecting diatom productivity, are commonly interpreted as a proxy for such upwelling –
with enhanced upwelling during southward displacement of the SHW (Anderson et al., 2009). Here we
only show records from the Pacific sector, given we find WDC $^{86}Kr_{xs}$ to reflect purely local SPJ dynamics
(Fig. 4A). Both published records suggest enhanced upwelling during the deglaciation (Fig. 7G), consistent
with a southward-shifted Pacific SPJ and enhanced storminess at WDC. The record from core PS75/072-4



(blue curve) further indicates an increasing productivity trend through the Holocene (Studer et al., 2018),
which is accompanied by a rise in surface nitrogen availability (reconstructed from diatom-bound nitrogen
isotopic composition, not shown); this Holocene trend matches our finding of increasing WDC storminess
and, by inference, an increasingly southern position of the Pacific SPJ and SHW. We thus conclude that
our interpretation of WDC $^{86}Kr_{xs}$ reflecting SPJ movement in parallel with the ITCZ, is broadly consistent
with indicators of wind-driven upwelling in the Pacific Antarctic zone.



## 6 Conclusions

Here we present and calibrate a new gas-phase ice core climate proxy, Kr-86 excess, that reflects time-averaged surface pressure variability at the site driven by synoptic activity. Surface pressure variability weakly disturbs the gravitational settling and enrichment of the noble gas isotope ratios $\delta^{86}Kr$ and $\delta^{40}Ar$ via barometric pumping. Owing to its higher diffusion coefficient, argon is less affected by this process than krypton is, and therefore the difference $\delta^{86}Kr$-$\delta^{40}Ar$ is a measure of synoptic activity.

This interpretation is supported by a calibration study in which we measure $^{86}Kr_{xs}$ in late Holocene ice core samples from eleven Antarctic and one Greenland ice core that represent a wide range of synoptic activity in the modern climate. Two of the Antarctic cores were rejected due to clear evidence of refrozen melt water. We find a strong correlation ($r = -0.94$ when using site mean data and $r = -0.83$ when using individual samples, $p < 0.01$) between ice core $^{86}Kr_{xs}$ and barometric variability at the site, demonstrating the validity of the new proxy.

Current limitations of the new $^{86}Kr_{xs}$ proxy are: (1) it requires relatively large and non-trivial corrections for gas loss and thermal fractionation; (2) it is moderately sensitive to changes in convective zone thickness; (3) firn air transport models cannot simulate the magnitude of $^{86}Kr_{xs}$ anomalies measured in ice samples; (4) firn air samples show smaller $^{86}Kr_{xs}$ anomalies than ice samples from the same site do; (5) it may be sensitive to the degree of density layering at the site, as a comparison of the nearby Law Dome DE08 and DSSW20K cores suggests; (6) it does not work for warm sites that experience frequent melt; (7) the measurement is challenging (with offsets observed between measurement campaigns), time consuming, and needs large ice samples; and (8) long-term sample storage may impose data offsets.

Using atmospheric reanalysis data, we show that synoptic-scale barometric variability in Antarctica is primarily linked to the position and/or strength of the southern edge of the eddy-driven subpolar jet (SPJ, also called polar front jet) with a southward SPJ displacement enhancing synoptic-scale surface pressure variability in Antarctica. The commonly-defined modes of large-scale atmospheric variability, such as the southern annular mode and the Pacific-South American pattern, impact Antarctic only weakly as they are weighted towards the mid-latitudes; the exception is the Antarctic Peninsula, where synoptic activity is well-correlated with the southern annular mode ($r = 0.68$). Sites in the Amundsen and Ross Sea sectors are weakly linked to tropical Pacific climate and ENSO ($r = 0.31$ to $r = 0.43$).

We present a new record of $^{86}Kr_{xs}$ from the WAIS Divide ice core in West Antarctica, that covers the last 24ka including the LGM, deglaciation and Holocene. West Antarctic synoptic activity is slightly below modern levels during the last glacial maximum (LGM); increases during the Heinrich Stadial 1 and Younger Dryas North Atlantic cold periods; weakens abruptly at the Holocene onset; remains low during the early and mid-Holocene (up to ~17% below modern), and gradually increases to its modern value. The WDC $^{86}Kr_{xs}$ record resembles records of tropical hydrology and monsoon intensity that are commonly thought to reflect the meridional position of the ITCZ; the sense of the correlation is that WDC synoptic activity is weak when the ITCZ is in its northward position, and vice versa. We interpret the record to reflect migrations of the eddy-driven SPJ in parallel with those of the ITCZ (Ceppi et al., 2013). Secondary influences may come from tropical Pacific climate and ENSO activity. Our $^{86}Kr_{xs}$ record is consistent with weakened ENSO activity (or a more La Niña-like mean state) during the mid- and early Holocene, and enhanced ENSO activity during NH stadial periods – both these features have been described in the paleo-



ENSO literature. The inferred changes to the SPJ are broadly consistent with proxies that indicate enhanced
wind-driven upwelling in the Pacific Antarctic zone during NH cold stadial periods.
Kr-86 excess is a new and potentially useful ice core proxy with the ability to enhance our understanding
of past atmospheric circulation. More work to better understand this proxy is warranted, and presently the
conclusions of this paper should be considered as tentative. In particular, replication of the deglacial Kr-86
excess record presented here in nearby cores is needed before these results can be interpreted with
confidence. Despite the many challenges of Kr-86 excess, its further development is worthwhile owing to
the dearth of available proxies for reconstructing SH extratropical atmospheric circulation.



## Appendix A: data corrections

### A1 Gas loss correction

Gas loss processes artificially enrich the $\delta^{40}Ar$ isotopic ratio used to calculate $^{86}Kr_{xs}$ (Kobashi et al., 2008b; Severinghaus et al., 2009; Severinghaus et al., 2003). Figure A1B shows the relationships between the two most common gas loss proxies $\delta O_2/N_2$ and $\delta Ar/N_2$ for all samples in the calibration dataset; we find a slope close to the 2:1 slope commonly reported in the literature (Bender et al., 1995); the exception is the DE08-OH site where the data fall on a 1:1 slope. Depletion in fugitive gases (such as $O_2$ and Ar) represents the sum of losses during bubble closure in the firn (Bender, 2002; Huber et al., 2006; Severinghaus and Battle, 2006), and those during drilling, handling, storage, and analysis of the samples (Ikeda-Fukazawa et al., 2005). The patterns are inconsistent with storage conditions alone – for example the DF and EDC cores were stored very cold and SP drilled very recently; yet all three have strong $\delta O_2/N_2$ and $\delta Ar/N_2$ depletion. Natural gas loss from the firn, as well as artefactual loss during drilling likely dominate the signal. The DE08-OH samples were dry-drilled and suffered from poor ice quality for the most depleted samples, which may explain the alternate 1:1 slope at the site (Appendix B); note though that a recent work suggests a ~5:1 slope for post-coring gas loss (Oyabu et al., 2021).

Severinghaus et al. (2009) hypothesize that the apparent 2:1 slope of $\delta O_2/N_2$ to $\delta Ar/N_2$ depletion is a combination of two mechanisms: size-dependent fractionation during diffusion through the ice lattice, and mass-dependent fractionation (such as molecular or Knudsen diffusion) within ice fractures. In this interpretation, the exact slope would depend on the relative contribution of each process to the total gas loss. It is improbable that both processes would occur in the same ratio at such a wide variety of sites; the 2:1 slope is thus more likely an attribute of the gas diffusion rate of gases through ice itself, which is strongly size-dependent, and weakly mass-dependent (Battle et al., 2011).

Gas loss is well known to enrich ice samples in $\delta^{18}O$-$O_2$, and following Severinghaus et al. (2009) we plot $\delta^{18}O$ (corrected for gravity and small atmospheric $\delta^{18}O_{atm}$ variations) against gravitationally-corrected $\delta O_2/N_2$ in Fig. A1C. We find a slope of 3.5 per meg enrichment in $\delta^{18}O$ per ‰ of $\delta O_2/N_2$ gas loss. This is less than values reported elsewhere (Severinghaus et al., 2009), but provides further evidence for mass-dependent fractionation during gas loss. Our core top dataset further suggests a correlation between $\delta^{40}Ar$ - $4 \times \delta^{15}N$ (a measure of $\delta^{40}Ar$ enrichment impacted by both thermal fractionation and gas loss) and gravitationally corrected $\delta Ar/N_2$ (Fig. A1D), suggesting Ar loss leads to enrichment of the remaining $\delta^{40}Ar$.

Following Severinghaus et al. (2009), we assume that the $\delta^{40}Ar$ correction scales with gas loss indicator ($\delta O_2/N_2$ - $\delta Ar/N_2$):

$$\Delta_{GL}^{40} = \varepsilon_{40} \times (\delta O_2/N_2 \text{ - } \delta Ar/N_2)|_{gravcorr} \quad\quad (A1)$$

with $\Delta_{GL}^{40}$ the isotopic gas loss correction on $\delta^{40}Ar$ and $\varepsilon_{40}$ a scaling parameter. Note that gravitationally corrected $\delta O_2/N_2$ and $\delta Ar/N_2$ data are used. Here we rely on data from the Antarctic Byrd ice core for a best estimate of $\varepsilon_{40}$ (Fig. A2); some samples from this core suffered extreme gas loss with ($\delta O_2/N_2$ - $\delta Ar/N_2$) as low as -100‰. This data set suggest $\varepsilon_{40}$ = -0.008, or 8 per meg $\delta^{40}Ar$ enrichment per ‰ of ($\delta O_2/N_2$ - $\delta Ar/N_2$) gas loss. Because of the 2:1 slope between $\delta O_2/N_2$ and $\delta Ar/N_2$, we find that ($\delta O_2/N_2$ - $\delta Ar/N_2$) ≈ $\delta Ar/N_2$



and therefore the coefficient $\varepsilon_{40}$ would have a similar slope when regressed against $\delta Ar/N_2$ instead of
$(\delta O_2/N_2 - \delta Ar/N_2)$.
The value of $\varepsilon_{40} = -0.008$ agrees reasonably well with other studies. Kobashi et al. (2008) compare replicate
sample pairs to back out gas loss, and find (statistically significant) correlations between $\delta^{40}Ar$ enrichment
and $\delta Ar/N_2$ (again, which is similar to $\delta O_2/N_2 - \delta Ar/N_2$). Kobashi et al. (2008) find $\varepsilon_{40}$ values of -0.006, -
0.005 and +0.007, depending on the depth range and analytical campaign evaluated. The positive value is
surprising, given that most observations, as well as theory, suggest $\varepsilon_{40}$ should be negative – we consider
this a spurious result given the weak $\delta^{40}Ar$ - $\delta Ar/N_2$ correlation in that particular data set. The other two
values of $\varepsilon_{40}$ are in reasonable agreement with the Byrd value. For the Siple Dome ice core (Severinghaus
et al., 2003), regressing $\delta^{40}Ar$ against $\delta Kr/Ar$ gives a slope of +0.007; this implies $\varepsilon_{40} = -0.007$ in good
agreement with our findings. Last, our coretop data suggest $\delta^{40}Ar$ enrichment with an $\varepsilon_{40}$ value of -0.0072
(Fig. A1D), also in good agreement with Byrd.
Given the uncertainty in the gas loss parameter, we verify that our results are valid for a wide range of $\varepsilon_{40}$
values (Fig. 3B).
**A2 Thermal correction**
In the presence of a temperature gradient, thermal diffusion causes isotopic enrichment towards the colder
location. The thermal diffusion sensitivity $\Omega$ in units of $‰K^{-1}$ for the various gases is given as (Grachev
and Severinghaus, 2003a, b; Kawamura et al., 2013):
$$\Omega^{15} = \frac{8.656}{T} - \frac{1232}{T^2}$$

$$\Omega^{40} = \frac{26.08}{T} - \frac{3952}{T^2}$$

$$\Omega^{86} = \frac{5.05}{T} - \frac{580}{T^2}$$

We estimate the thermal gradient $\Delta T$ in the firn using N-15 excess (Severinghaus et al., 1998):
$$\Delta T = \frac{^{15}N_{xs}}{\Omega^{15} - \Omega^{40}/4} = \frac{\delta^{15}N - (\delta^{40}Ar + \Delta_{GL}^{40})/4}{\Omega^{15} - \Omega^{40}/4} \qquad (A2)$$

with $\Delta_{GL}^{40}$ the $\delta^{40}Ar$ gas loss correction from Eq. (A1). Positive values of $\Delta T$ indicate that the surface is
warmer than the firn-ice transition. The $\Delta T$ then in turn allows us to estimate the thermal corrections:
$$\Delta_{TF}^{15} = -\Omega^{15}\Delta T$$

$$\Delta_{TF}^{40} = -\Omega^{40}\Delta T$$

$$\Delta_{TF}^{86} = -\Omega^{86}\Delta T \qquad (A3)$$



The samples from the calibration dataset are from the climatically stable late Holocene period, and typically
close together in depth; the uncertainty in the $\Delta T$ estimation for individual samples therefore exceeds the
temporal variability in $\Delta T$. To reduce the uncertainty in the thermal correction we estimate $\Delta T$ for individual
samples using Eq. (A2), and for each site average the available data to get a site-average firn temperature
gradient $\overline{\Delta T}$. The thermal correction is then given by:
$$\Delta_{TF}^{15} = -\Omega^{15}\overline{\Delta T}$$
$$\Delta_{TF}^{40} = -\Omega^{40}\overline{\Delta T}$$
$$\Delta_{TF}^{86} = -\Omega^{86}\overline{\Delta T} \qquad\qquad (A4)$$
The two methods are compared in Figs. A3C (individual sample $\Delta T$) and A3D (site mean $\overline{\Delta T}$); it is clear
that the $\overline{\Delta T}$ approach reduces the spread in $^{86}Kr_{xs}$ (error bars), but not its mean (white dots). The $\Delta T$ estimates
in individual samples are subject to errors in the isotopic measurements; some of these errors will cancel
out in the $\overline{\Delta T}$.
For the downcore WDC record through the deglaciation we can no longer assume a stationary $\Delta T$; we
instead rely on dynamic firn densification model simulations of $\Delta T$ (Buizert et al., 2015). A comparison of
the simulated and data-based $\Delta T$ is shown in Fig. A5 for WDC. The data clearly show a lot more
scatter/variability than the simulations do. We interpret this mainly as analytical noise in the $\delta^{15}N$ and $\delta^{40}Ar$
measurements, however, the gas loss correction (Appendix A1) also impacts the $\Delta T$ estimation in individual
samples. The comparison suggests that the scatter in the $\Delta T$ estimates actually exceeds the magnitude of
the simulated thermal signals. Using $\Delta T$ of the individual samples would thus introduce much scatter in the
(thermally corrected) $^{86}Kr_{xs}$ records, and we choose to use the modelled $\Delta T$ instead.
**A3 Elevation correction**
To correct the deglacial WAIS Divide record for elevation changes, we here estimate the $^{86}Kr_{xs}$ dependence
on site elevation using the calibration dataset. Note that elevation and synoptic activity are strongly
correlated for the investigated sites ($r = -0.86$), with synoptic activity decreasing with elevation because
the cyclonic systems do not penetrate deeply into the Antarctic interior. Figure A6 shows the result of this
exercise. We find a slope of 34 per meg ‰ of $^{86}Kr_{xs}$ per 1000 m of elevation change, with a correlation of
$r = 0.96$ when considering site-mean $^{86}Kr_{xs}$, and $r = 0.86$ when considering individual samples. Note that
the GISP2 site is not included in the analysis because it is in Greenland where the elevation-$^{86}Kr_{xs}$
relationship may be different from Antarctica – it does however fit the Antarctic trend rather well. We
further use the simulated WAIS Divide elevation history (Golledge et al., 2014), which simulates an LGM
elevation of around 300m higher than at present at WAIS Divide.





## Appendix B: Sub-annual $^{86}Kr_{xs}$ variations at DE08-OH

The Law Dome DE08-OH site is a revisit of the DE08 site, drilled in the 2018/2019 Austral summer Antarctic field season. We have samples from two separate cores: (1) thirteen 24-cm-long samples from a 10-cm-diameter core going from 97 m to 193 m depth at ~ 8 m sample spacing; and (2) eight 6-cm-long samples from a 24-cm-diameter core going from 97.6 m to 99.8 m depth at 30 cm sample spacing. The purpose of the first set was to determine possible long-term variations in $^{86}Kr_{xs}$; the purpose of the second set to assess whether there are sub-annual variations in $^{86}Kr_{xs}$ due to the seasonality in firn properties and bubble trapping.

Both cores were dry-drilled (i.e., no drill liquid was used). The 10-cm-diameter core used was drilled at the beginning of the field season, the 24-cm-diameter core at the end of the field season. Prior to shipment off the continent, both cores were stored in a chest freezer at Casey Station; due to a miscommunication this freezer was set to -20ºC rather than -26ºC, yet the ice is believed to have stayed below -18ºC.

Both DE08-OH cores experienced more gas loss than the original DE08 core that we also sampled (Fig. A1 B). In particular the samples from the 10-cm-diameter core were strongly depleted in $\delta Ar/N_2$, with the most extreme gas loss seen for the deepest samples where the ice quality was poorest.

Fig. B1 shows the high-resolution sub-annual DE08-OH sampling. The data were corrected for gas loss and thermal fractionation, using a site-mean temperature gradient of $\overline{\Delta T}$ = -1.6ºC, possibly related to a rectifier effect (Morgan et al. 2022). We find strong (5-fold) variations in $^{86}Kr_{xs}$ on sub-annual time scales. With an expected annual layer thickness of around 1.3 m at this depth, it appears as though there may be an annual-scale variation in $^{86}Kr_{xs}$; the data set has insufficient length to establish this firmly.

We refrain from interpreting the long-term variations in $^{86}Kr_{xs}$ in the 10-cm-diameter core for two reasons. First, given the strong sub-annual variations seen in the high-resolution sampling, it is unavoidable that we are aliasing the underlying signal in the core. Second, the 10-cm-diameter core suffers from strong gas loss (depleted $\delta Ar/N_2$). We attribute this primarily to the dry drilling and imperfect sample storage conditions. Perhaps the greater stresses during drilling a 10-cm core (compared to the 24-cm diameter core) result in more micro-fractures and gas loss.



**Supplement**

A data supplement is available with this paper.

**Data availability**

Data are available here: https://www.usap-dc.org/view/project/p0010037, and via the data supplement to this paper.

**Author contributions**

CB, JS, AJS and EJB designed research; SS, AS, BB, KK, DB, AJS, JDM and IO contributed measurements; KK, DME, NB, RLP, RB, EM-T, PDN, DT, and VVP contributed ice core samples; CB and WHGR analyzed reanalysis data; CB, AJS, and BB performed firn modelling; CB drafted the manuscript with input from all authors.

**Competing Interests**

The authors declare no competing interests.

**Acknowledgements**

The idea for the Kr-86 excess proxy came out of discussions at the 2014 WAIS Divide Ice Core Science Meeting held at Scripps Institution of Oceanography in La Jolla, CA. The authors want to thank John Chiang, Justin Wettstein, Zanna Chase, Bob Anderson, Tyler Jones and Eric Steig for useful discussions, data sharing and manuscript feedback, the NSF ice core facility (NSF-ICF, formerly the National Ice Core Laboratory) for curating and distributing ice core samples, the European Centre for Medium-Range Weather Forecasts (ECMWF) for making ERA-Interim reanalysis datasets publicly available, and the US ice drilling program for coordinating ice core drilling in Antarctica. Sample collection at Law Dome was supported by the Australian Antarctic Science Program, the Australian Antarctic Division and (at DE08-OH) the U.S. National Science Foundation.

**Financial Support**

We gratefully acknowledge financial support from the U.S. National Science Foundation (grant numbers ANT-0944343, ANT-1543267, ANT-1543229, ANT-1643716 and ANT-1643669), the New Zealand Ministry of Business, Innovation and Employment (grant numbers RDF-VUW-1103, 15-VUW-131, 540GCT32).





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



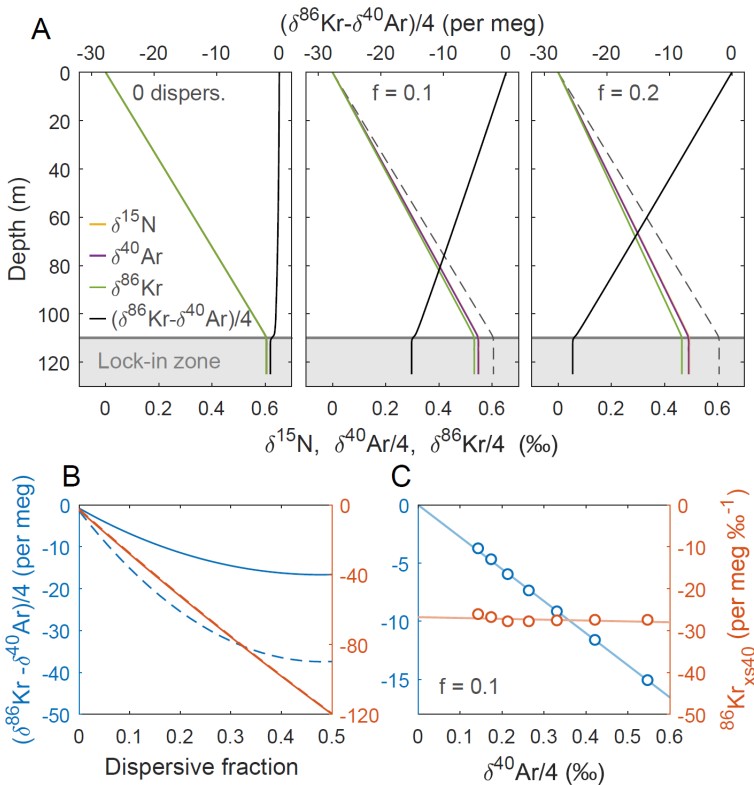

**Figure 1.** Idealized firn air transport model experiments of $^{86}Kr_{xs}$. Firn density is calculated using (Herron and Langway, 1980), and the diffusivity using (Schwander, 1989). **A** Simulations using a fraction of dispersive mixing of $f = 0$ (left), $f = 0.1$ (middle) and $f = 0.2$ (right) for a hypothetical site with accumulation rate of $A = 2$ cm a$^{-1}$ ice equivalent and mean annual temperature $T = -60$ºC. At dispersive fraction $f$, effective molecular diffusivity of all gases is multiplied by $(1-f)$ and dispersive mixing for all gases is set equal to $f$ times the effective molecular diffusivity of $CO_2$. **B** Isotopic disequilibrium as a function of dispersive mixing intensity at two different firn thicknesses of around 100 m (dashed, $A = 2$ cm a$^{-1}$ and $T = -60$ºC) and 50 m (solid, $A = 2$ cm a$^{-1}$ and $T = -43$ºC). We compare isotopic disequilibrium without (blue, left axis) and with (orange, right axis) normalization. **C** Simulations at 10 % dispersive mixing, where each dot represents different climatic conditions. Accumulation rate is $A = 2$ cm a$^{-1}$ ice equivalent and mean annual temperature is changed from -60ºC to -30ºC in steps of 5ºC.





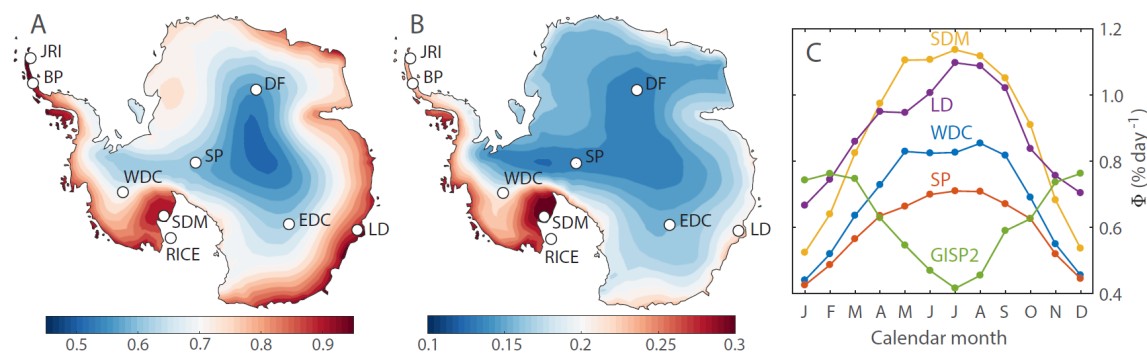

1200

1201

**Figure 2.** Calibrating Kr-86 excess. **A** Annual-mean $\Phi$ in Antarctica over 1979-2017, in units of % day$^{-1}$.
**B** Interannual variability (1$\sigma$ standard deviation) of annual-mean $\Phi$ over 1979-2017, in units of % day$^{-1}$. **C**
Annual cycle in $\Phi$ for 1979-2017 for the indicated sites.

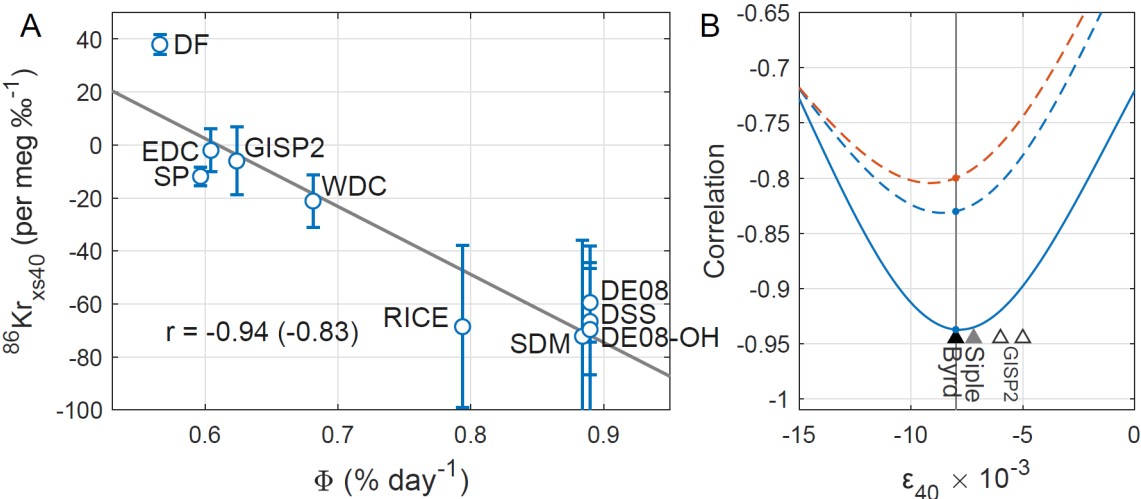

**Figure 3.** Calibrating Kr-86 excess. **A** $^{86}Kr_{xs}$ as a function of $\Phi$ for the calibration data set. Circles give the site mean, and the error bars denote the $\pm 1\sigma$ standard deviation between samples (uncertainty in corrections not included). Pearson correlation coefficient is $r = -0.94$ when considering site data means and $r = -0.83$ when considering all individual samples. Data are corrected for gas loss using $\varepsilon_{40} = -0.008$ (Appendix A1), and corrected for thermal fractionation using site-mean N-15 excess (Appendix A2). The calibration curve for $^{86}Kr_{xs15}$ is identical in this case, with slightly larger errorbars. **B** Correlation of the calibration curve as a function of the gas loss correction scaling parameter $\varepsilon_{40}$. The solid line gives the correlation for both site-mean $^{86}Kr_{xs15}$ and $^{86}Kr_{xs40}$ (identical); the dashed lines the correlation using individual samples for $^{86}Kr_{xs40}$ (blue) and $^{86}Kr_{xs15}$ (orange). Triangles denote the $\varepsilon_{40}$ estimate from the Byrd, Siple and GISP2 ice cores (Fig. A2; Kobashi et al., 2008a; Severinghaus et al., 2003).



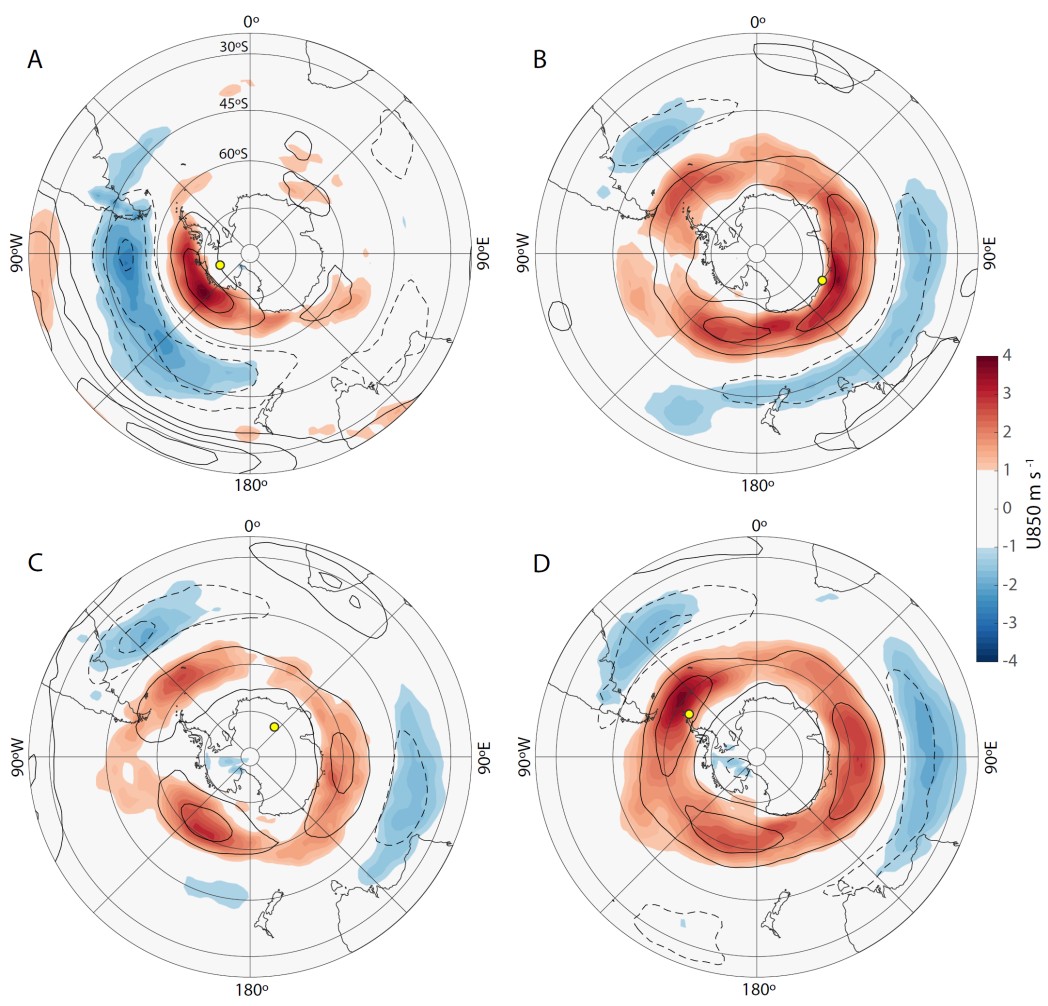

1216

**Figure 4.** Zonal wind speed at 850 hPa (color shading, see scale bar) and 200 hPa (2 m s$^{-1}$ contours) regressed onto surface synoptic activity Φ at the Antarctic ice core sites of: **A** WAIS Divide; **B** Law Dome (DE08, DE08-OH and DSSW20K); **C** Dome Fuji; **D** James Ross Island. Yellow dots mark the ice core locations.

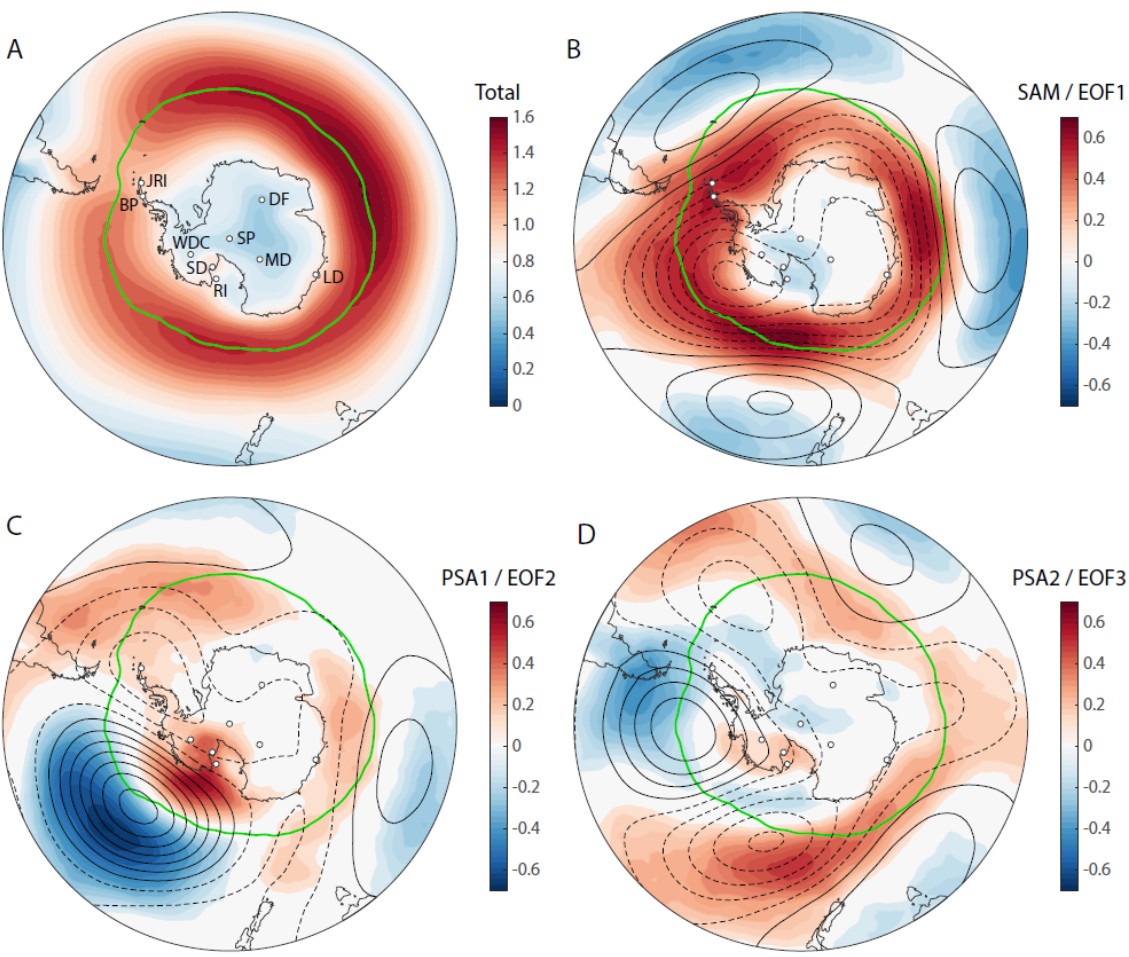

**Figure 5.** Modes of SH extratropical atmospheric variability and their link to synoptic-scale surface pressure variability in Antarctica. **A** Annual mean Φ in units of % day$^{-1}$; latitude of maximum Φ denoted by green line. **B** Colors show correlation between Φ and the Southern Annular Mode (SAM) index, with superimposed the 500 hPa geopotential height anomalies in 10 m contours. **C** as panel B, but for the Pacific-South American Pattern 1 (PSA1). **D** As panel B, but for the Pacific-South American Pattern 2 (PSA2). SAM, PSA1 and PSA2 are defined as respectively the first, second and third EOFs (Empirical Orthogonal Functions) of the 500 hPa geopotential height anomalies in 20°-90°S monthly values in the 1979-2017 ERA interim reanalysis (Dee et al., 2011).

1230

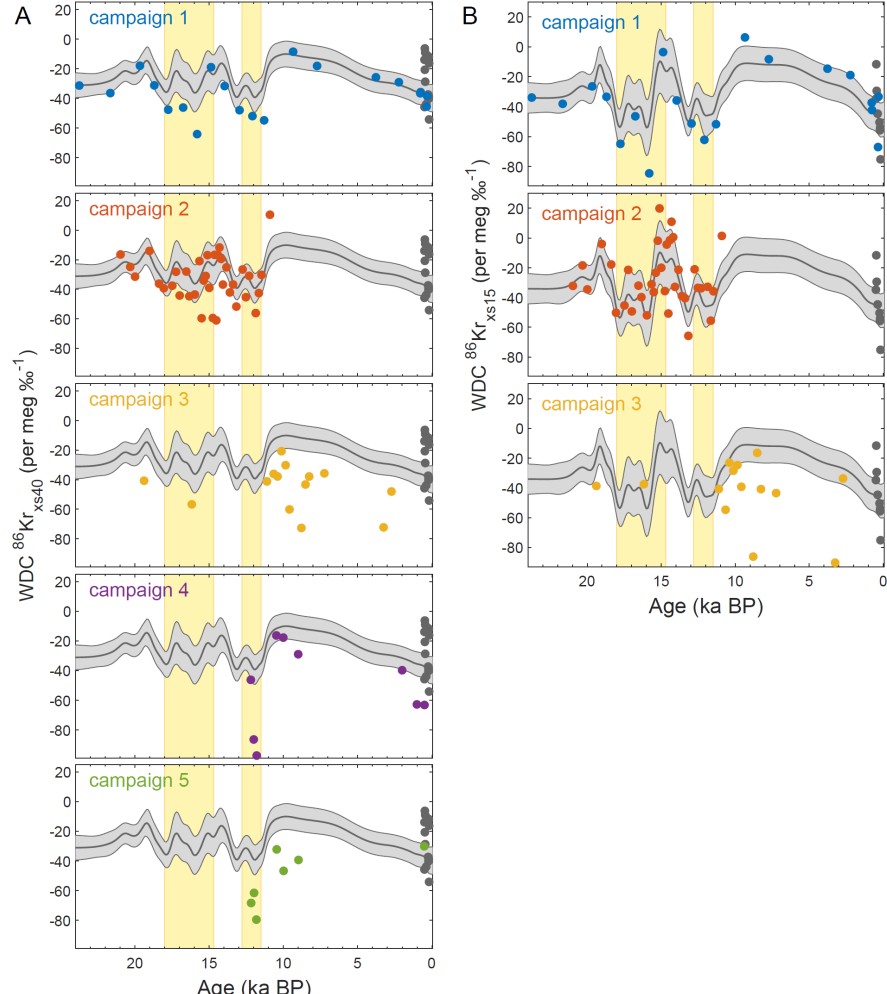

1231

1232

**Figure 6.** WAIS Divide Kr-86 excess records through the last deglaciation. **A** WDC $^{86}Kr_{xs40}$ data from the five measurement campaigns. The gray curve shows a Gaussian smoothing curve to the combined data from the first two campaigns; the light gray shaded area shows the $\pm 1\sigma$ uncertainty envelope based on a 10,000 iteration Monte-Carlo sampling of the errors and uncertainties. The WDC calibration data is shown as gray circles for comparison. **B** As in panel (A), but for $^{86}Kr_{xs15}$. For campaigns 4 and 5 the sample was not split, and no $\delta^{15}N$ data are available. The Heinrich Stadial 1 and Younger Dryas North-Atlantic cold periods marked in yellow.





1240

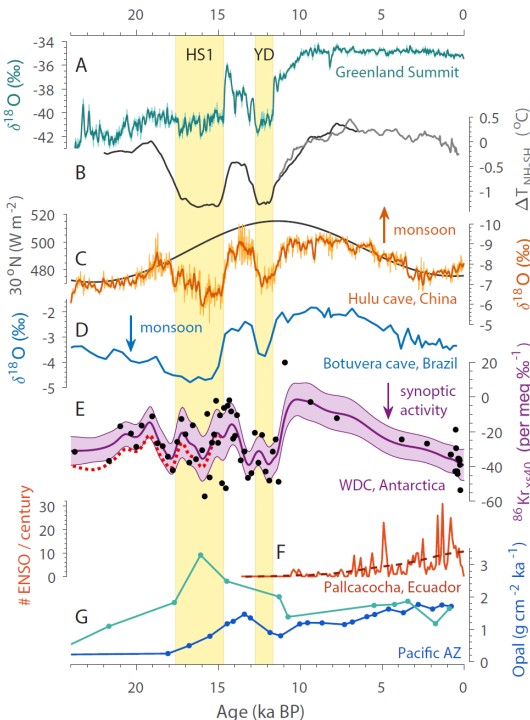

1241

**Figure 7.** Climate records through the last deglaciation with the Heinrich Stadial 1 (HS1) and Younger
Dryas (YD) North-Atlantic cold periods marked in yellow. **A** Greenland Summit ice core stable water
isotope ratio $\delta^{18}$O here the average of the GISP2 and GRIP ice cores (Grootes et al., 1993). **B** Hemispheric
temperature difference (McGee et al., 2014) based on global proxy compilations for the Holocene (Marcott
et al., 2013) and last deglaciation (Shakun et al., 2012). **C** Speleothem calcite $\delta^{18}$O from Hulu and Dongge
Caves, China, as a proxy for East Asian summer monsoon strength (Dykoski et al., 2005; Wang et al.,
2001). Superimposed is summer solstice (June 21) insolation at 30ºN. **D** Speleothem calcite $\delta^{18}$O from
Botuvera cave, southern Brazil, as a proxy for South American summer monsoon strength (Cruz et al.,
2005; Wang et al., 2007). **E** Kr-86 excess record from WAIS Divide (this study); corrected for gas loss and
thermal fractionation (Appendix A). Center line and shaded envelope show the mean and ±1σ uncertainty
interval of a 10,000 iteration Monte Carlo smoothing exercise (see text). The dotted red line equals the
center line with a correction for elevation change applied (Appendix A) using a simulated elevation history
(Golledge et al., 2014). **F** Number of El Niño events per century from laminations in sediments from Laguna
Pallcacocha, Ecuador (Moy et al., 2002). **G** Th-normalized opal flux in the Pacific Antarctic zone (south of
the polar front) from cores NBP9802-6PC1 (turquoise; 169.98ºW, 61.88ºS) and PS75/072-4 (blue;
151.22ºW, 57.56ºS), reflecting local productivity and (wind-driven) upwelling (Chase et al., 2003; Studer
et al., 2015). All isotope data in this figure are on the V-SMOW scale. Arrows show direction of increased
monsoon strength / synoptic activity.

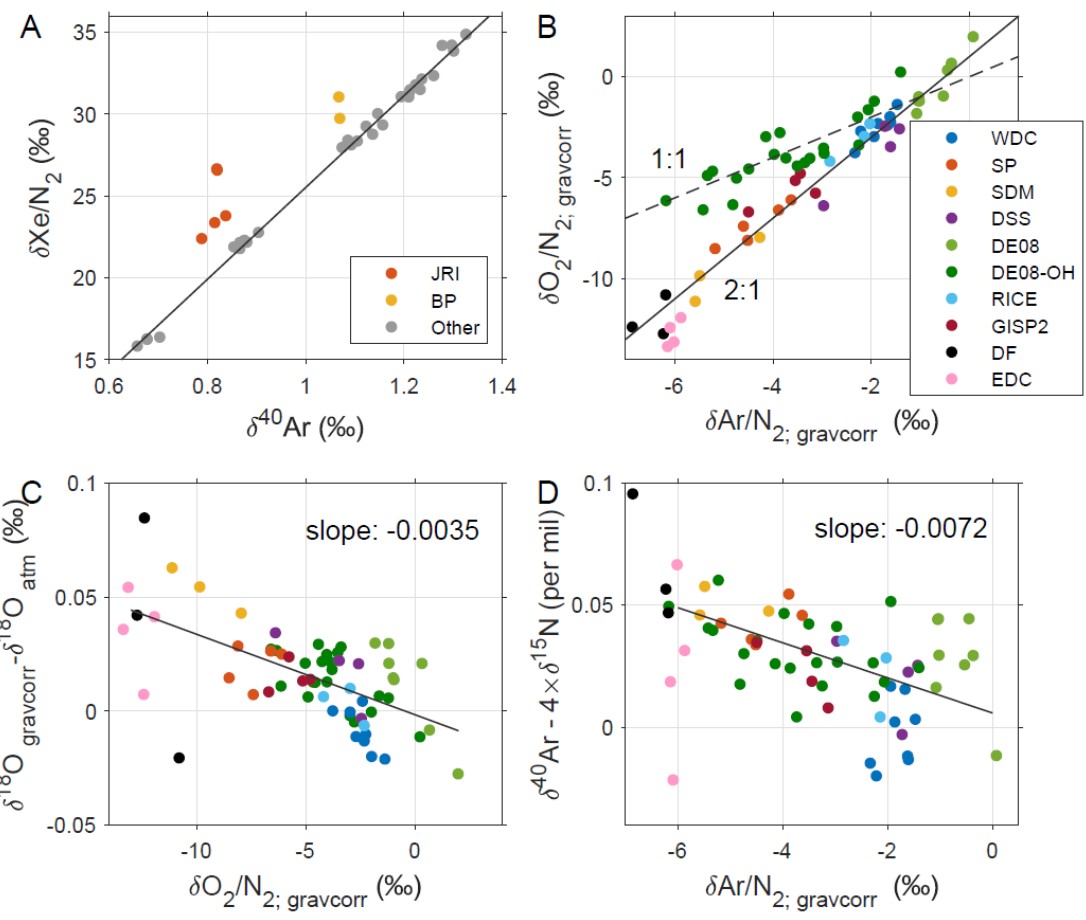

1260

**Figure A1.** Elemental ratios in the 11-site calibration study of late Holocene samples. **A** $\delta Xe/N_2$ vs. $\delta^{40}Ar$ in all ice core samples. $\delta^{40}Ar$ is used solely to illustrate gravitational enrichment, and a similar picture arises when plotted against any isotopic pair. Refrozen meltwater (elevated $\delta Xe/N_2$) was seen in all samples from the Antarctic Peninsula (James Ross Island and Bruce Plateau sites), despite selecting samples free of visible melt features. **B** The relationship between the commonly used gas loss proxies $\delta O_2/N_2$ and $\delta Ar/N_2$ corrected for gravity. **C** Enrichment in $\delta^{18}O$ (corrected for gravity and atmospheric $\delta^{18}O_{atm}$) plotted against gravity-corrected $\delta O_2/N_2$ **D** $\delta^{40}Ar$ enrichment plotted against gravity-corrected $\delta Ar/N_2$. In all panels gravitational correction is applied by subtracting $\delta^{15}N$ times the atomic mass unit difference.



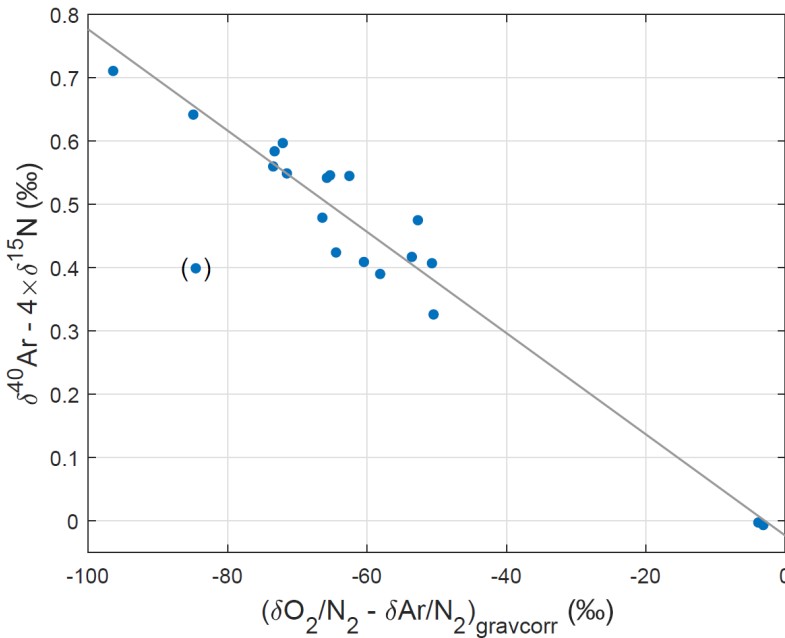

1269

**Figure A2.** Argon isotopic enrichment due to gas loss. The enrichment in $\delta^{40}$Ar plotted as a function of gravitationally corrected ($\delta O_2/N_2$ - $\delta Ar/N_2$) measured in the deep Antarctic Byrd ice core, which suffered heavy gas loss. Ice samples were analyzed in the Bender Lab at the University of Rhode Island by Jeff Severinghaus in 1997. The slope of the least-square fit is $\varepsilon_{40}$ = -0.008. The data point in parentheses is treated as an outlier and excluded from the fitting.

1275

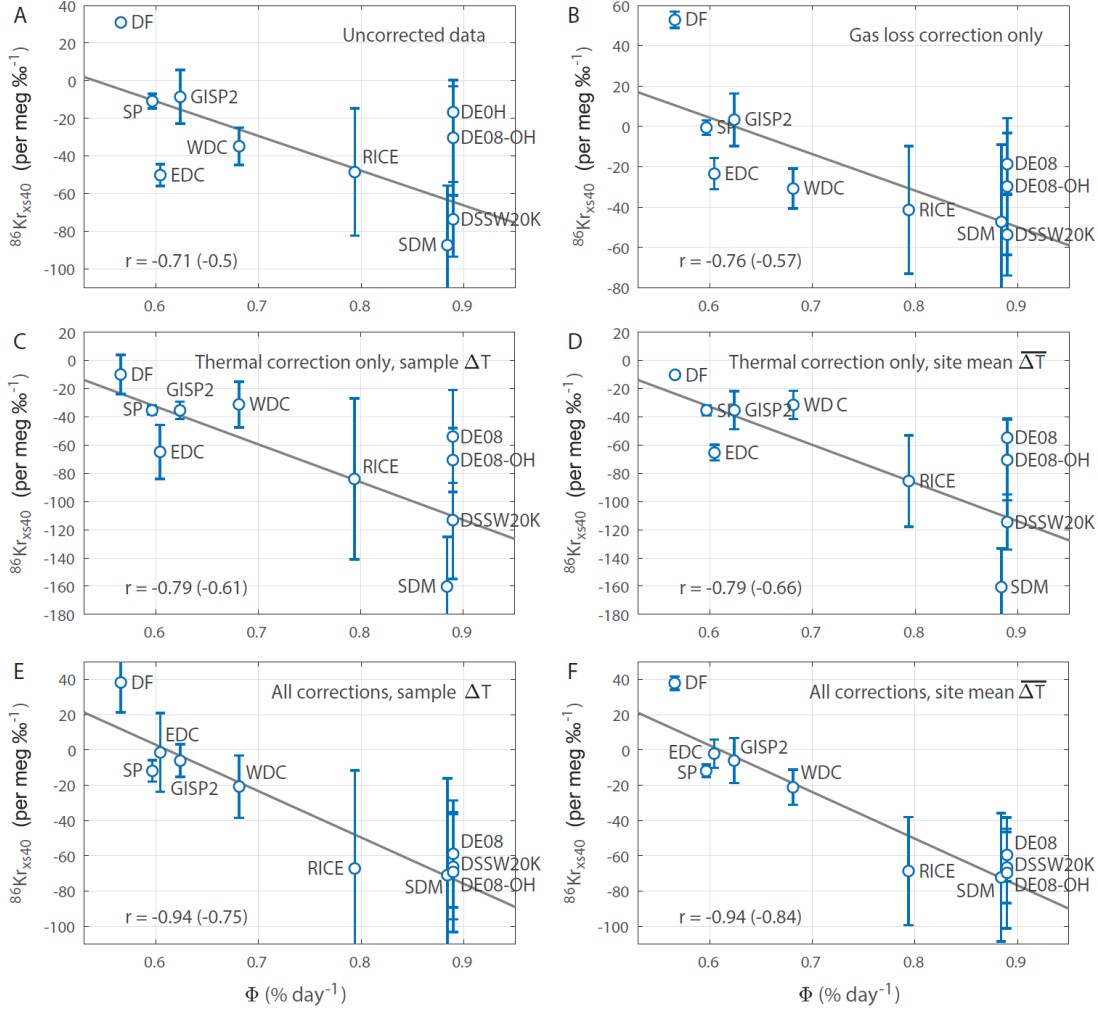

1276

1277

**Figure A3.** Influence of gas loss and thermal correction on the $^{86}Kr_{xs40}$ calibration. We plot $^{86}Kr_{xs40}$ as a function of $\Phi$ **A** without any data corrections applied; **B** with only the gas loss correction applied ($\varepsilon_{40}$ = -0.008); **C** with only the thermal correction applied using individual sample $\Delta T$; **D** with only the thermal correction applied using individual site mean $\overline{\Delta T}$; **E** with both gas loss and thermal corrections applied using individual sample $\Delta T$; **F** with both gas loss and thermal corrections applied using site mean $\overline{\Delta T}$. In each panel the correlation to $\Phi$ are listed for the site-average and individual sample with the latter in parentheses. For all correlations $p < 0.05$.



**Figure A4.** Same as figure A3, but for $^{86}Kr_{xs15}$. Note that the gas loss correction (panel **B**) does not impact $^{86}Kr_{xs15}$. For all correlations $p < 0.05$, except for panels **A** and **B** where $p = 0.16$ for the site-average correlation.





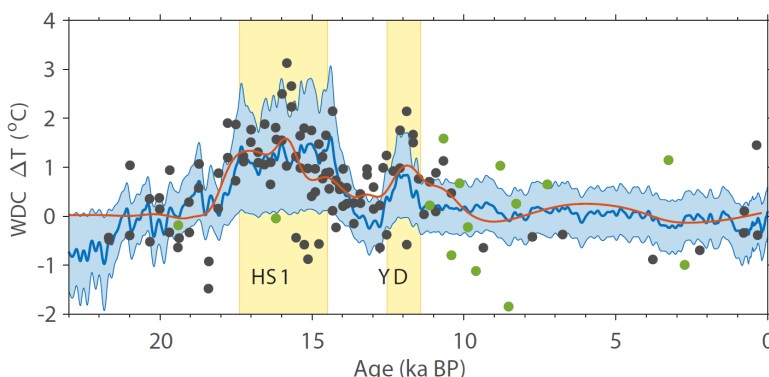


**Figure A5.** The $\Delta T$ correction applied to the downcore records. Blue envelope shows the $\pm 2\sigma$ range of thermal correction scenarios in the Monte Carlo sampling, together with the mean (blue line). Gray dots show WDC $\Delta T$ estimates from available $^{15}$N-excess data, with the red curve being a Gaussian smoothing function to the data. Green dots are $^{15}$N-excess from campaign 3, showing somewhat greater scatter.



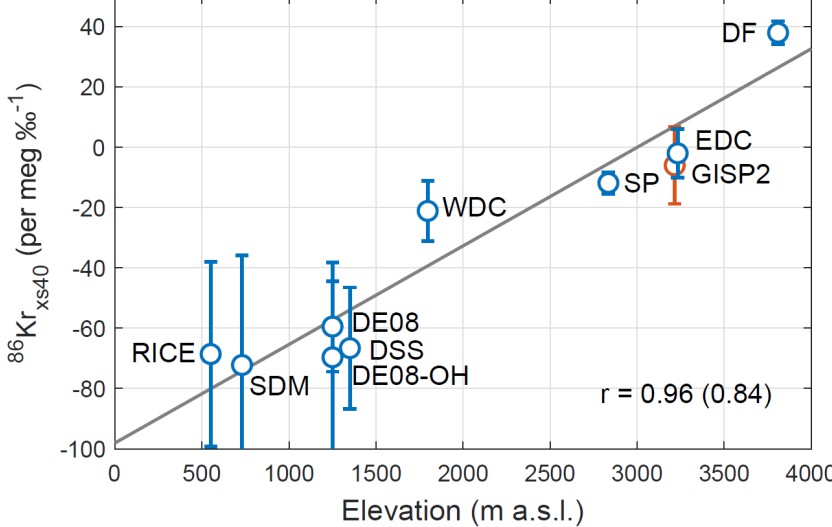


**Figure A6.** Kr-86 excess dependence on site elevation. Vertical axis is the $^{86}$Kr$_{xs}$. The linear fit has a slope of 34 per meg ‰$^{-1}$ per 1000 m elevation.






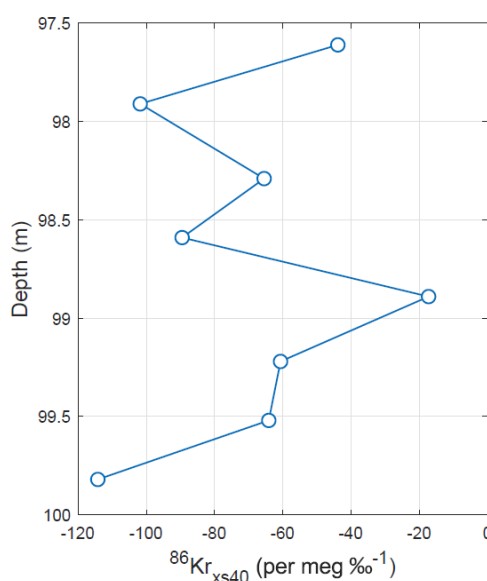


**Figure B1** High-resolution sub-annual sampling of $^{86}Kr_{xs40}$ in the DE08-OH site. The annual layer thickness
at this depth is around 1.3 m.




**Table 1.** Ice core sites used in this study, with *N* the number of samples included in the calibration study. See the main text for acronyms.

| Site | $T$ (°C) | $A$ (m ice a$^{-1}$) | $\Phi$ (% day$^{-1}$) | Latitude | Longitude | $N$ |
|------|------|------|------|------|------|------|
| WDC | -31 | 0.22 | 0.68 | 79.5°S | 112.1°W | 8[a] |
| DF | -57 | 0.028 | 0.56 | 77.3°S | 39.7°E | 3 |
| SP | -51 | 0.078 | 0.6 | 90.0°S | 98.2°W | 5 |
| SDM | -25 | 0.13 | 0.88 | 81.7°S | 149.1°W | 3 |
| DSSW20K | -21 | 0.16 | 0.89 | 66.8°S | 112.6°E | 4 |
| DE08 | -19 | 1.2 | 0.89 | 66.7°S | 113.2°E | 8 |
| DE08-OH | -19 | 1.2 | 0.89 | 66.7°S | 113.2°E | 8[b] |
| RICE | -24 | 0.24 | 0.79 | 79.4°S | 161.7°W | 3[a] |
| EDC | -55 | 0.03 | 0.6 | 75.1°S | 123.4°E | 4 |
| JRI | -14 | 0.68 | 0.97 | 64.2°S | 57.7°W | 5[c] |
| BP | -15 | 2 | 0.9 | 66.1°S | 64.1°W | 2[c] |
| GISP2 | -32 | 0.23 | 0.62 | 72.6°N | 38.5°W | 4 |

[a] Not including one sample rejected due to technical problems.
[b] Only shallow samples due to strong gas loss in deeper samples attributed to warm storage conditions.
[c] Refrozen meltwater present as indicated by elevated Xe/N$_2$ ratio.

**Table 2.** Pearson correlation between $\Phi$ at the ice coring sites and large-scale atmospheric circulation. Correlations are calculated using annual mean data (all months, April-March). We only list the statistically significant correlations ($p < 0.1$). The Niño 3.4 is calculated over 5°S - 5°N, 190°E - 240°E, using SST from Huang et al. (2014); the PDO index is from Mantua and Hare (2002).

| Site | SAM | PSA1 | PSA2 | Niño 3.4 | PDO | Sea ice Am-Bell | Sea ice Ross |
|------|------|------|------|------|------|------|------|
| WDC | - | 0.31 | - | 0.31 | 0.28 | - | - |
| SDM | - | 0.47 | 0.34 | 0.43 | 0.45 | - | -0.32 |
| RICE | - | 0.41 | 0.34 | 0.34 | 0.45 | - | -0.30 |
| SP | - | - | -0.32 | - | -0.30 | - | - |
| LD | 0.45 | - | - | - | - | - | - |
| DF | 0.37 | - | - | - | - | - | - |
| EDC | 0.30 | - | - | - | - | - | - |
| JRI | 0.67 | - | - | - | - | 0.31 | - |
| BP | 0.68 | - | - | - | - | - | - |