# Peer review of "The new Kr-86 excess ice core proxy for synoptic activity: West Antarctic"

_Climate of the Past, 2022_

## Referee Comment (RC1)

Review of "The new Kr-86 excess ice core proxy for synoptic activity: West Antarctic storminess possibly linked to ITCZ movement through the last deglaciation." by Christo Buizert et al, Climate of the Past.

**General:**

The manuscript presents a new, exiting interpretation of older and younger ice core elemental and isotope ratio results. I enjoyed reading it. The results of combined proxy, called $^{86}Kr_{ex40}$, are based on relative difference of $\delta^{86}Kr$ ($^{86}Kr/^{82}Kr$ ratio values) and $\delta^{40}Ar$ ($^{40}Ar/^{36}Ar$ ratio values) to $\delta^{40}Ar$. It thus corresponds to a delta value of delta values expressed in permeg/permil when the primary delta values are expressed in permil. This relative double difference results in very small values and are therefore they expressed in permeg, which is a permil of permil. The measurements show that the values corrected for thermal diffusion are in the small negative range of 0 to -160 permeg/permil for $^{86}Kr_{ex40}$. The authors state that $^{86}Kr_{ex40}$ is a direct proxy of large-scale atmospheric circulation (synoptic-scale pressure variability). Yet, they are careful with their interpretation as there are still insufficient knowledge of the underlying firn air transport and gas trapping which may influence $^{86}Kr_{ex40}$. There are a couple of major points that should be addressed before the manuscript can be published.

Major points:
(1) Figure 3 is one of the major Figures and these values depend on two corrections applied (gas loss and thermal diffusion) which are detailed in Figure A3. Looking at the supplementary Figure A3 that displays the uncorrected and corrected values for gas loss and thermal diffusion independently, I saw that there must be an interdependence of these two corrections as they are not adding up. For instance for DF the uncorrected mean value is around 33 permeg/permil, the gas loss corrected about 55 permeg/permil which leads to a gas loss correction of around 22 permeg/permil. The correspondent thermal correction amounts to (33-(-10) = -43 permeg/permil both for individual or mean $\Delta T$. This in combination would lead to correction of -21 permeg/permil (-43+22). I therefore would except overall corrected value of around 12 permeg/permil (33-21). The values plotted are, however close to +40 permeg/permil? Could you explain how and why they are interlinked, or is there a mistake for the DF values? The other site value corrections are more or less additive, maybe with the additional exception of GISP2.
Actually, the same issue concerns Figure A4.

(2) There is hardly any information/discussion about the many more elemental and isotope ratio measurements that have been measured (section 2.2) to strengthen or weaken their arguments, i.e. $^{84}Kr/^{86}Kr$, $^{84}Kr$ being the most abundant and therefore the precision should be better.

(3) The expression of $^{86}Kr_{ex40}$ being a direct proxy for synoptic-scale pressure variability comes at several places and is actually quite misleading as they correctly state that the gas measurements represent a time-averaged value. The average times are large (years to decades) compared to synoptic circulation events (days).

(4) Calibration of the $^{86}Kr_{ex40}$ has been done with reanalysis data of the time range 1979 to 2017. This data show a large spatial variability in the Antarctic. However, whether the stability of the spatial calibration will hold for temporal interpretations is difficult to judge but

this is certainly one major weakness. Yet, I see that it will be difficult to find arguments to support it.

(5) In section 1.2, the authors discuss several processes that alter the isotope ratios, such as gravitational settling and thermal diffusion, advection, convection and dispersive mixing. The latter three they state do not distinguish between isotopologues. This is correct but they do lead to a disruption of the maybe already established isotope equilibrium through molecular (gravitational and/or thermal) diffusion, which requires time to be re-established. This is the starting point of their definition of the $^{86}Kr_{ex40}$ proxy. However, there are several processes that can and will affect this proxy as nicely discussed in sections 1.2 and 3.3. This is of course a weakness of this proxy despite the author's transparent writing. For instance, they state that the major influence on $^{86}Kr_{ex40}$ comes from pressure variations at the surface, but what about the pressure variations from the gas close-off process? Pressure variations may be weak but gas velocities of expelled molecules in the tiny channels at close-off depths might be very high and could lead to significant alterations in the gas compositions.

Minor points:

L144:     it would be worthwhile to explain why this definition is less sensitive to thermal diffusion (give corresponding reference). Yet,

L144ff:   What about close-off fractionation? We know that Ne, He, will expelled during close-off. Therefore, large molecules such as $N_2$ will be less affected than $O_2$, Ar etc. Kr is obviously between $N_2$ and $O_2$. Especially, Ar will be subject of expelling. In this regard, the second definition with $\delta^{15}N$ would be preferred.

L225ff:   I would prefer the gas splitting. As it can be tested by many gas species measurements in contrast to the different ice core samples. There only replications can help.

L239f     How extended is this bubble-clathrate zone as the signals are extremely small. What was the criteria for the given number in the depth or time range.

L242ff    "Some of the EDC samples analyzed had clear evidence of drill liquid contamination, which acts to artefactually lower $^{86}Kr_{ex40}$; the late Holocene data used here were not flagged for drill liquid contamination." Give a reference for this statement.

L247      22 per meg /permil: I do not understand this, error propagation leads to a higher combined uncertainty and not a lower!, this does not make sense or do I miss something here?

L248      this would be necessary. At least you can split a 1600 g samples in two sub-samples

L253      BP also denotes Before Present, consider changing it.

L257ff    Are there additional indications that melting has occurred, for instance from water isotopes or changed greenhouse gas concentrations?

L268      Can you specify what modern climate means (time range)

L274  How can a daily variable be compared to a decadal variable ($^{86}Kr_{ex40}$)?

L278f  If such a calibration is made, it should be done on firn air samples as they are not smoothed by the process of gas enclosure. Have you tried to do this?

L288f  This is only the case when the argon correction for gas loss has been made correctly.

L299ff  Refer to Figure 3.

L304ff  This is disconnected, the link is not clear. Further explanation is needed here.

L310ff  This is indeed a critical point.

L332f  This is again a critical point as I can imagine that the firn structure acts as a column retarding the gas species differently. It would be worthwhile to do such experiments. Maybe you find a corresponding reference?

L346ff  This is also important. Elemental ratio should be in line with isotope ratios. Yet it points indeed to a difference in diffusion coefficient ratio of real and lab conditions. Column effect (adsorption/desorption)

L361f  if one argues that the diffusion of noble gases may be retarded in the firn column, one should consider this effect also for the thermal corrections. These, however is based on $\delta^{15}N$ and $\delta^{40}Ar$ measurements.

L368ff  this is also a very critical correction as obliviously $\Delta T$ varies considerably from site to site without a clear understanding why this is the case.

L378  give a reference for (1)

L379  (3) yes, this indicates the large uncertainty of this correction. Yet, Figure 3B is quite convincing as a counter-argument. See also main comments above.

L 410  How has the elevation changed over the course of the investigated period? And how relevant are these changes?

L416  …(by limiting …), not clear, needs further explanation

L429f  …we anticipate $^{86}Kr_{ex40}$ to be a qualitative proxy for synoptic variability…
    this is indeed a good point as the used calibration is standing on weak grounds.

L456ff  Why have you only investigated Antarctic sites and not Greenland locations? This would proof that different locations on Earth would be similarly influenced. There is GISP2. What would be a good choice for additional stations in Greenland?

L549f  or is there indeed a higher variability present. How do you explain or underline that less care has been taken for these later campaigns?

L575f  Campaign 3 data shows quite a large scatter.

L600ff  this might be tackled with measurements in Greenland compared to those in Antarctica

L702ff   Such a shift is not …after accounting for site elevation effects
this again is quite a critical point. This depends on ice flow modeling and
accumulation rates changes.

L730ff   But a similarly strong correlation is seen with mean annual site T and with site
elevation.

L756   …and gradually increases …
maybe due to low number of data points?

L773f   Gas loss correction,
this is a critical but important correction which still sits on shaky grounds.

L803f   Why do you define this correction like this? There is also a very good correlation
for mass spectrometry measurements. If the MS measurements are not well done
you might introduce a wrong correction.

Fig. 2   It would be worthwhile to plot also the temporal variability. This would allow the
reader to compare the uncertainty of annual means and to compare it with the
seasonality.

Fig. 3   Over which time range are the circles mean taken? Is this firn air or ice core data?
B: Sensitivity study. This is nice. It would be worthwhile to show a similar sensitivity
study for $^{86}Kr_{ex}$ for $\delta^{15}N$ based. $N_2$ is believed not to be influenced from gas loss!!

Fig. 6   The variability of the grey circles (calibration data) is as large as the variations of
all other measurements. Hence, the interpretation seems to be quite speculative.
In particular, that also other process could have caused similar variations
(convective zone changes).

---

## Community Comment (CC1)

**Comments on "The new Kr-86 excess ice core proxy for synoptic activity: West Antarctic storminess possibly linked to ITCZ movement through the last deglaciation"**

**Community comments by Aymeric P. M. Servettaz**

Japan Agency for Marine-Earth Science and Technology, Yokosuka, 237-0061, Japan

Preprint text cited in this comment is cited in orange color.

**Several occurrences:** WDC $^{86}Kr_{xs}$ is sometimes noted WD $^{86}Kr_{xs}$.

**Page 2 Line 40-41:** The abstract should precise whether subpolar jet of Northern Hemisphere or Southern Hemisphere is discussed, as both Greenland and Antarctica are previously mentioned.

**Page 5 Line 138 (and 142):** in the equations $^{86}Kr_{xs}$ is written as the difference to a thermally corrected $\delta^{40}Ar_{corr}$ (or $\delta^{15}N_{corr}$). While this mirrors the deviation from the gravitational fractionation, it covers the fact that both $\delta^{40}Ar$ and $\delta^{15}N$ are used in the thermally corrected data. I agree that this expression emphasizes the pumping-induced deviation from a gravitational settling, but it should be noted in the main text that three pairs of isotopes are necessary to express the $^{86}Kr_{xs}$.

Mathematically, the notation "per meg ‰$^{-1}$" could be simplified to "‰" and requires clarification. From my understanding the authors want to emphasize that it is normalized by the gravitational fractionation. Perhaps, a mention to why "per meg ‰$^{-1}$" is used should be given along with the "The rationale for including a normalization in the denominator is discussed below." (**Line 140**)

**Page 5 Line 146:** Fig. 6 is called before any other figure

**Page 11 Lines 346-348:** Authors write "Firn models predict that the gravitational disequilibrium effect in elemental ratios (such as δKr/Ar) should be proportional to that in isotopic ratios. However, the observations suggest that the former is usually smaller than would be expected from the latter. We do not have an explanation for this effect." Does the same reasoning that was done for the $^{86}Kr_{xs}$ apply for elemental ratios? The authors have written just above "krypton is more readily adsorbed onto firn surfaces retarding its movement" (**Page 10 line 333**). Retarding krypton movement could lower its effective diffusive column height, leading to lower gravitational enrichment relative to other elements.

I also have a more open question: Could we theoretically compute a Kr-excess equivalent derived from elemental ratio of Kr/Ar rather than the isotopic ratios of $^{86}Kr/^{82}Kr$, supposing we can discriminate between elemental gas loss from pore closure and elemental ratio changes from the active mixing of firn gases (in addition to thermal and gravitational effects)?

**Page 12 Line 389:** the authors write: "the gas age distribution at the depth of bubble closure has a width of several years" to discard the influence of sub-annual variations on Kr isotopes. The Kr-86 excess is used as a proxy for deviation from the gravitational equilibrium, which can be seen as "an effective diffusive column height" (**Line 385**). Although the gravitational equilibrium is indeed reach after several years as the gases go through the entire diffusive column height (DCH), I would suppose deviation from this equilibrium may be achieved within much shorter periods, because it relies on kinetic mixing. Then
we need to better understand where the Kr-excess signal is acquired. If the entire firn air column is
actively pumped out and pushed back in due to the passage of depression system, my guess would be
that the kinetic motion would affect the gases depending on the diffusivity in the column (or the inverse
of porosity). Could it imprint a new Kr-excess signal directly into the deep firn layers, even if the gases
have been effectively isolated from the atmosphere for a longer period and have an age distribution of
several years? Or is the entirety of Kr-excess signal acquired at the top of diffusive column through
exchanges with the open atmosphere?

**Page 14 Line 477:** "The green line denotes the latitude of maximum Φ, corresponding roughly to the
latitude with the highest storm track density (57.8°S on average)." Should be in the figure 5 caption.
Also, in the text "°S" is written with a superscript "o" letter in lieu of a degree sign.

**Page 19 Lines 654-657:** The authors write "the present-day SAM does not have a statistically significant
impact on synoptic variability at WDC (Table 2). Perhaps the SAM is not a good analogue for these past
changes in circulation after all, in particular when considering the impact of SHW shifts on Antarctic
storminess" to question the fact that "present-day SAM is sometimes suggested as an analogue for past
shifts in the meridional position of the SHW and eddy-driven jet" (**line 650**).

I think this justification is not logical. Here, the authors show in their Fig. 5B that the correlation
between storminess and SAM is limited to the oceanic regions, and is only weakly correlated on the
coastal regions of the Antarctic continent (except a high correlation in the marine-dominated Antarctic
Peninsula). This is supported by other studies showing that positive SAM is associated with more
frequent cyclones (Grieger et al., 2018), and their location is shifted south but limited to the oceanic
regions around Antarctica, with limited impact inland (Pezza et al., 2008). I do agree with the later
statement that "synoptic activity at WDC is not sensitive to the SAM" (**line 664**) and this may be true for
other sites inland Antarctica.

This does not impede the relation between SAM and westerlies, because the of SAM signature on
pressure variability may be restricted to a narrow band of latitudes where SAM-related changes on
storm activity is located (north of ~70°S). Modelling and reanalysis studies show that there are clear
connections between the SAM phase and the surface SHW strength and position (Marshall and
Thompson, 2016), or between SAM and the polar and subtropical jets (Fogt and Marshall, 2020).
Confusion may arise from the fact that southward shifts of SHW as reported from Fig. 4A influence the
Φ value at WDC, which clearly shows "the impact of SHW shifts on Antarctic storminess" (**line 657**).
However, this pattern of wind changes is zonally asymmetric, and resembles more changes associated
with the PSA1 as shown in Fig. 5C, with a geopotential high anomaly in the Pacific. Pressure variability
(Φ) at WDC may therefore be driven by changes in PSA1. In my understanding this is a complex situation
where changes in westerlies related to SAM variability do not influence the storminess at WDC, but
other changes in westerlies (mainly PSA1?) may change the storminess at WDC.

I would like to add that even though pressure variability at WDC is not influenced by SAM, some other
parameters such as source water for precipitations (as recorded in deuterium excess) are influenced by
SAM and may reflect more zonally symmetric changes (Markle et al., 2017; Buizert et al., 2018). Direct
comparison of the two proxies in a future study may prove interesting, and here in this study
interpretations of Kr-86 excess from WDC should rely more on the geographical extent of the regression
shown in Fig. 4A.

**Page 26 Line 862:** "per meg ‰" is missing an exponent (‰$^{-1}$)

**Page 26 Line 867:** missing a space in "300m"

**Page 38 Line 1222:** The contour lines lack a description to discriminate between positive geopotential
height anomalies (continuous lines) and negative anomalies (dashed lines).

**Page 39 Line 1238:** it is noted that "For campaigns 4 and 5 the sample was not split, and no $\delta^{15}$N data
are available". It is unclear if the thermal correction for $\delta^{40}$Ar was calculated in these campaigns, as
Appendix A2 mentions the need for $^{15}$N$_{xs}$ in this correction.

**References cited in this document**

Buizert, C., Sigl, M., Severi, M., Markle, B. R., Wettstein, J. J., McConnell, J. R., Pedro, J. B., Sodemann, H.,
Goto-Azuma, K., Kawamura, K., Fujita, S., Motoyama, H., Hirabayashi, M., Uemura, R., Stenni, B.,
Parrenin, F., He, F., Fudge, T. J., and Steig, E. J.: Abrupt ice-age shifts in southern westerly winds
and Antarctic climate forced from the north, Nature, 563, 681–685,
https://doi.org/10.1038/s41586-018-0727-5, 2018.
Fogt, R. L. and Marshall, G. J.: The Southern Annular Mode: Variability, trends, and climate impacts
across the Southern Hemisphere, WIREs Clim Change, 11, https://doi.org/10.1002/wcc.652,
2020.
Grieger, J., Leckebusch, G. C., Raible, C. C., Rudeva, I., and Simmonds, I.: Subantarctic cyclones identified
by 14 tracking methods, and their role for moisture transports into the continent, Tellus A:
Dynamic Meteorology and Oceanography, 70, 1–18,
https://doi.org/10.1080/16000870.2018.1454808, 2018.
Markle, B. R., Steig, E. J., Buizert, C., Schoenemann, S. W., Bitz, C. M., Fudge, T. J., Pedro, J. B., Ding, Q.,
Jones, T. R., White, J. W. C., and Sowers, T.: Global atmospheric teleconnections during
Dansgaard–Oeschger events, Nature Geoscience, 10, 36–40, https://doi.org/10.1038/ngeo2848,
2017.
Marshall, G. J. and Thompson, D. W. J.: The signatures of large-scale patterns of atmospheric variability
in Antarctic surface temperatures: Antarctic Temperatures, J. Geophys. Res. Atmos., 121, 3276–
3289, https://doi.org/10.1002/2015JD024665, 2016.
Pezza, A. B., Durrant, T., Simmonds, I., and Smith, I.: Southern Hemisphere Synoptic Behavior in Extreme
Phases of SAM, ENSO, Sea Ice Extent, and Southern Australia Rainfall, Journal of Climate, 21,
5566–5584, https://doi.org/10.1175/2008JCLI2128.1, 2008.

---

## Author Comment (AC1)

**Comments by Aymeric Servettaz.**

We want to thank Aymeric for his thoughtful and constructive comments that have improved the manuscript. Our responses below. We wrote our responses in the form of proposed changes to the text that we would make in a potential revised manuscript.

WDC 86 Krxs is sometimes noted WD 86Krxs

Thanks for catching. We now use WDC throughout, and have replaced three instances of "WD".

Page 2 Line 40-41: The abstract should precise whether subpolar jet of Northern Hemisphere or Southern Hemisphere is discussed, as both Greenland and Antarctica are previously mentioned.

Good point. We added "Antarctic" $^{86}$Kr$_{xs}$ and "southern hemisphere".

Page 5 Line 138 (and 142): in the equations 86Krxs is written as the difference to a thermally corrected δ40Arcorr (or δ15Ncorr). While this mirrors the deviation from the gravitational fractionation, it covers the fact that both δ40Ar and δ15 N are used in the thermally corrected data. I agree that this expression emphasizes the pumping-induced deviation from a gravitational settling, but it should be noted in the main text that three pairs of isotopes are necessary to express the 86Krxs.

We have added the following to the text (below Eq 3):

Note that both definitions rely on having measurements of three isotope ratios ($\delta^{86}$Kr, $\delta^{40}$Ar and $\delta^{15}$N), as the thermal correction requires $\delta^{40}$Ar and $\delta^{15}$N be known

Mathematically, the notation "per meg ‰$^{-1}$" could be simplified to "‰" and requires clarification. From my understanding the authors want to emphasize that it is normalized by the gravitational fractionation. Perhaps, a mention to why "per meg ‰$^{-1}$" is used should be given along with the "The rationale for including a normalization in the denominator is discussed below." (Line 140)

Agreed. We added:

This unit (per meg ‰$^{-1}$) is mathematically identical to ‰, but we use it to emphasize the normalization in the denominator.

Page 5 Line 146: Fig. 6 is called before any other figure

We have removed this reference to Fig. 6

Page 11 Lines 346-348: Authors write "Firn models predict that the gravitational disequilibrium effect in elemental ratios (such as δKr/Ar) should be proportional to that in isotopic ratios. However, the observations suggest that the former is usually smaller than would be expected from the latter. We do not have an explanation for this effect." Does the same reasoning that was done for the 86Krxs apply for elemental ratios? The authors have written just above "krypton is more readily adsorbed onto firn surfaces retarding its movement" (Page 10 line 333). Retarding krypton movement could lower its effective diffusive column height, leading to lower gravitational enrichment relative to other elements

Yes, adsorption could perhaps explain the observation in case there is no isotopic fractionation associated with this process. We added:

As before, adsorption of Kr onto firn grain surfaces may contribute to the observed discrepancy, and laboratory tests of this process are called for. Further, the impacts of gas loss are greater on elemental ratios than on the isotopic ratios which may contribute also.

I also have a more open question: Could we theoretically compute a Kr-excess equivalent derived from elemental ratio of Kr/Ar rather than the isotopic ratios of 86Kr/82Kr, supposing we can discriminate between elemental gas loss from pore closure and elemental ratio changes from the active mixing of firn gases (in addition to thermal and gravitational effects)?

This is an interesting question, and in theory it would be possible. The isotopic definition is preferred for multiple reasons:

 First of all, measuring isotopic ratios is more precise than measuring elemental ratios. For isotope ratios, all masses are monitored simultaneously in the IRMS on different cups at a single magnet setting. For elemental ratios we have to rely on so-called "peak jumping", where the IRMS magnet is switched. This means that isotope ratios can be measured to a greater precision that elemental ratios can.

Second, argon suffers from gas loss in ice core samples, which impacts the Kr/Ar ratio more than the $^{40}Ar/^{36}Ar$ ratio. This complicates a definition of disequilibrium based on elemental ratios. Besides the artifactual gas loss from samples in storage, there is indeed the natural size fractionation during close-off that the reviewer refers to that one would also have to account for.

Last, the definition of disequilibrium requires a comparison between two gravitational ratios. A disequilibrium proxy based on elemental ratios would for example compare the Kr/Ar ratio to the Ar/N2 ratio. There would be a smaller contrast in diffusion rates, which makes the proxy less sensitive.

Page 12 Line 389: the authors write: "the gas age distribution at the depth of bubble closure has a width of several years" to discard the influence of sub-annual variations on Kr isotopes. The Kr-86 excess is used as a proxy for deviation from the gravitational equilibrium, which can be seen as "an effective diffusive column height" (Line 385). Although the gravitational equilibrium is indeed reach after several years as the gases go through the entire diffusive column height (DCH), I would suppose deviation from this equilibrium may be achieved within much shorter periods, because it relies on kinetic mixing. Then we need to better understand where the Kr-excess signal is acquired. If the entire firn air column is actively pumped out and pushed back in due to the passage of depression system, my guess would be that the kinetic motion would affect the gases depending on the diffusivity in the column (or the inverse of porosity). Could it imprint a new Kr-excess signal directly into the deep firn layers, even if the gases have been effectively isolated from the atmosphere for a longer period and have an age distribution of several years? Or is the entirety of Kr-excess signal acquired at the top of diffusive column through exchanges with the open atmosphere?

This is a very good question that goes to the heart of the current difficulty in interpreting this proxy. We fully agree with the reviewer sentiment that we need to better understand where within the firn column the Kr-excess signal is acquired. If the pores that facilitate the barometric pumping flow represent a small fraction of the entire firn cross-section, then indeed a single barometric pumping event could conceivably introduce a large amount of unfractionated air deep into the firn, and impact the atmospheric composition close to pore close-off. But more likely, the barometric pumping displaces air in the deep firn over much shorter distances, providing a longer integration time of the barometric

signal. In response to this comment, we modified the text. We now state that the seasonal variation in storminess as an explanation for the observations "seems improbable to us at present". We indeed cannot rule out a scenario as sketched by the reviewer.

Page 14 Line 477: "The green line denotes the latitude of maximum Φ, corresponding roughly to the latitude with the highest storm track density (57.8°S on average)." Should be in the figure 5 caption. Also, in the text "°S" is written with a superscript "o" letter in lieu of a degree sign

This was mentioned in the caption under panel 5A, but we now specify that it holds for all panels. The use of a superscript "o" is out of laziness, as it is faster to type in MS word. We fixed this instance pointed out by the reviewer, but cannot rule out other instances in the text. We trust this will be fixed by the copy editor during typesetting.

Page 19 Lines 654-657: The authors write "the present-day SAM does not have a statistically significant impact on synoptic variability at WDC (Table 2). Perhaps the SAM is not a good analogue for these past changes in circulation after all, in particular when considering the impact of SHW shifts on Antarctic storminess" to question the fact that "present-day SAM is sometimes suggested as an analogue for past shifts in the meridional position of the SHW and eddy-driven jet" (line 650).

I think this justification is not logical. Here, the authors show in their Fig. 5B that the correlation between storminess and SAM is limited to the oceanic regions, and is only weakly correlated on the coastal regions of the Antarctic continent (except a high correlation in the marine-dominated Antarctic Peninsula). This is supported by other studies showing that positive SAM is associated with more frequent cyclones (Grieger et al., 2018), and their location is shifted south but limited to the oceanic regions around Antarctica, with limited impact inland (Pezza et al., 2008). I do agree with the later statement that "synoptic activity at WDC is not sensitive to the SAM" (line 664) and this may be true for other sites inland Antarctica.

This does not impede the relation between SAM and westerlies, because the of SAM signature on pressure variability may be restricted to a narrow band of latitudes where SAM-related changes on storm activity is located (north of ~70°S). Modelling and reanalysis studies show that there are clear connections between the SAM phase and the surface SHW strength and position (Marshall and Thompson, 2016), or between SAM and the polar and subtropical jets (Fogt and Marshall, 2020). Confusion may arise from the fact that southward shifts of SHW as reported from Fig. 4A influence the Φ value at WDC, which clearly shows "the impact of SHW shifts on Antarctic storminess" (line 657). However, this pattern of wind changes is zonally asymmetric, and resembles more changes associated with the PSA1 as shown in Fig. 5C, with a geopotential high anomaly in the Pacific. Pressure variability (Φ) at WDC may therefore be driven by changes in PSA1. In my understanding this is a complex situation where changes in westerlies related to SAM variability do not influence the storminess at WDC, but other changes in westerlies (mainly PSA1?) may change the storminess at WDC. I would like to add that even though pressure variability at WDC is not influenced by SAM, some other parameters such as source water for precipitations (as recorded in deuterium excess) are influenced by SAM and may reflect more zonally symmetric changes (Markle et al., 2017; Buizert et al., 2018). Direct comparison of the two proxies in a future study may prove interesting, and here in this study interpretations of Kr-86 excess from WDC should rely more on the geographical extent of the regression shown in Fig. 4A.

It is unclear to us what the reviewer is suggesting we do or change here in response to this comment.

The SAM is commonly defined in terms of sea level pressure and not in terms of the actual atmospheric dynamics. Sea level pressure integrates over dynamics in the entire atmospheric column. Our point is merely that the modern-day interannual variance in the SHW (the "SAM") is dominated by internal variability, whereas changes to the SHW on orbital timescale are driven by the energy distribution at the surface. Due to geostrophy any shift in the SHW, regardless of its origin or dynamics, will map onto the SAM index as conventionally defined. The SAM index by itself is simply not a good tool to discuss the *dynamics* of the SHW, particularly on longer timescales. Internal month-to-month SAM variability is expected to occur even when the mean position of the SHW were to be shifted due to a change in orbital configuration. It is also confusing that in common usage the SAM *index* and the internal mode of variability that leads to variations in the SAM index are both referred to simply as "the SAM".

In response to the reviewer comment we have rewritten the paragraph to make our point more clearly. It may have been confusing the start the paragraph noting that the SAM is an analogue for past shifts in the SHW – which it probably is not. We have changed this in the revised text. We have made further edits for clarity:

*"The SAM index reflects the meridional position of the SHW and eddy-driven jet. During positive SAM phases the SHW are displaced poleward, and during negative phases equatorward. Present-day month-to-month changes in SAM index represent a mode of internal variability, with anomalies persisting for only weeks to months – the timescale is longest in late spring and early summer reflecting a stronger planetary wave–mean flow interaction (Simpson et al., 2011; Thompson and Wallace, 2000). By contrast, shifts in the ITCZ and SH jet structure on millennial and orbital timescales have a much longer lifetime and different dynamics, being driven from the tropics via hemispherically asymmetric changes in Hadley cell and STJ strength. Therefore, present-day SAM internal variability is not expected to be a good analogue for past changes in SHW position. We find that the present-day SAM month-to-month internal variability mainly impacts synoptic variability over the Southern Ocean and does not have a statistically significant WDC (Table 2). Such variability is likely to have occurred during other climatic regimes also, possibly just centered around a mean SHW position that is displaced meridionally relative to today. At first glance it may appear contradictory to state, as we do, that synoptic activity at WDC is not sensitive to the SAM while also suggesting that during the last deglaciation synoptic activity at WDC is linked to changes in the position of the SH eddy-driven jet and westerlies. Based on the considerations above, both claims may be true without contradiction."*

Page 26 Line 862: "per meg ‰" is missing an exponent (‰-1)

Fixed!

Page 26 Line 867: missing a space in "300m"

Fixed!

Page 38 Line 1222: The contour lines lack a description to discriminate between positive geopotential height anomalies (continuous lines) and negative anomalies (dashed lines).

Thanks, we added this into the caption.

Page 39 Line 1238: it is noted that "For campaigns 4 and 5 the sample was not split, and no δ15N data

are available". It is unclear if the thermal correction for δ40Ar was calculated in these campaigns, as Appendix A2 mentions the need for 15Nxs in this correction.

Good point. We added:

*"Thermal corrections in the WDC $^{86}Kr_{xs}$ records are based on fin model simulations."*

---

## Author Comment (AC2)

**Comments by Reviewer #1**

We want to thank Reviewer #1 for their thoughtful and constructive comments that have improved the manuscript. Our responses below. We wrote our responses in the form of proposed changes to the text that we would make in a potential revised manuscript.

General: The manuscript presents a new, exiting interpretation of older and younger ice core elemental and isotope ratio results. I enjoyed reading it. The results of combined proxy, called 86Krex40, are based on relative difference of δ86Kr ( 86Kr/ 82Kr ratio values) and δ40Ar ( 40Ar/ 36Ar ratio values) to δ40Ar. It thus corresponds to a delta value of delta values expressed in permeg/permil when the primary delta values are expressed in permil. This relative double difference results in very small values and are therefore they expressed in permeg, which is a permil of permil. The measurements show that the values corrected for thermal diffusion are in the small negative range of 0 to -160 permeg/permil for 86Krex40. The authors state that 86Krex40 is a direct proxy of large-scale atmospheric circulation (synoptic-scale pressure variability). Yet, they are careful with their interpretation as there are still insufficient knowledge of the underlying firn air transport and gas trapping which may influence 86Kr ex40.

There are a couple of major points that should be addressed before the manuscript can be published.

Major points:

(1) Figure 3 is one of the major Figures and these values depend on two corrections applied (gas loss and thermal diffusion) which are detailed in Figure A3. Looking at the supplementary Figure A3 that displays the uncorrected and corrected values for gas loss and thermal diffusion independently, I saw that there must be an interdependence of these two corrections as they are not adding up. For instance for DF the uncorrected mean value is around 33 permeg/permil, the gas loss corrected about 55 permeg/permil which leads to a gas loss correction of around 22 permeg/permil. The correspondent thermal correction amounts to (33-(-10) = -43 permeg/permil both for individual or mean ΔT. This in combination would lead to correction of -21 permeg/permil (-43+22). I therefore would except overall corrected value of around 12 permeg/permil (33-21). The values plotted are, however close to +40 permeg/permil? Could you explain how and why they are interlinked, or is there a mistake for the DF values? The other site value corrections are more or less additive, maybe with the additional exception of GISP2.

Actually, the same issue concerns Figure A4.

We checked our calculations, and there is no mistake for the DF values. There is indeed an interdependence of these two corrections, and they are not additive. The observations of the reviewer are thus absolutely correct. The reason is that both involve the d40Ar isotopic ratio. The gas loss correction makes the d40Ar values smaller, which by itself makes $^{86}Kr_{xs}$ more positive at all sites. However, this change in the $\delta^{40}Ar$ also changes the estimated firn temperature gradient ΔT, because it is based on the 15N excess ($\delta^{15}N$-$\delta^{40}Ar$/4). Performing the thermal correction either with or without the gas loss correction therefore will give different results. There is therefore not a single value for the thermal correction, and the size of this correction is dependent on the size of the gas loss correction.

At none of the sites we expect the corrections to be exactly additive, however, depending on the details of the site they may appear approximately additive.

(2) There is hardly any information/discussion about the many more elemental and isotope ratio measurements that have been measured (section 2.2) to strengthen or weaken their arguments, i.e. 84Kr/ 86Kr, 84Kr being the most abundant and therefore the precision should be better.

Yes, $^{84}$Kr is indeed more abundant, resulting in the largest signal on the IRMS cup. However, it has a smaller mass difference with the other isotopes. The $\delta^{86/82}$Kr has the largest mass difference (4 mass units), and we find that of all the isotope pairs it typically has the best precision per unit mass difference. For this reason, and for the sake of consistency, we decided to use this isotope pair throughout for the krypton isotopes.

In her PhD thesis, Anais Orsi introduced the concept of $\delta$*Kr, which is the weighted average of all Krypton isotope pairs ($\delta^{86/82}$Kr/4, $\delta^{84/83}$Kr and $\delta^{86/84}$Kr/2), weighted by the standard deviation of the repeat measurements for each pair. For several of the sites we compared $\delta$*Kr and $\delta^{86}$Kr, and did not find much difference between these two. In future work we plan to investigate the difference more systematically.

(3) The expression of 86Kr ex40 being a direct proxy for synoptic-scale pressure variability comes at several places and is actually quite misleading as they correctly state that the gas measurements represent a time-averaged value. The average times are large (years to decades) compared to synoptic circulation events (days).

This is a good point. While we tried to make this point clearly in the text, it could easily be misunderstood. We considered using the term "pressure variance" instead of "pressure variability", but that has a mathematical meaning that is distinct from the way we define $\Phi$. To address this point, we have replaced these statements with "time-averaged pressure variability" instead, to clarify that we cannot resolve individual storm systems. We also added a statement to the abstract to reflect this:

*"The $^{86}Kr_{xs}$ reflects the time-averaged synoptic pressure variability over several years (site "storminess"), and does not record individual synoptic events."*

(4) Calibration of the 86 Kr ex40 has been done with reanalysis data of the time range 1979 to 2017. This data show a large spatial variability in the Antarctic. However, whether the stability of the spatial calibration will hold for temporal interpretations is difficult to judge but this is certainly one major weakness. Yet, I see that it will be difficult to find arguments to support it.

This is indeed an important point, and we agree that our approach provides only the first-order proof. This issue has long plagued the interpretation of other ice core proxies, most notably the $\delta^{18}$O of ice. Future efforts in climate modeling, combined with more observations of Kr-86 excess through time are needed to move beyond the spatial calibration. We address this point in the manuscript:

*"The calibration of the $^{86}Kr_{xs}$ proxy is based on spatial regression. In applying the proxy relationship to temporal records, we make the implicit assumption that proxy behavior in the temporal and spatial dimensions is at least qualitatively similar. This assumption may prove incorrect. In particular, changes in insolation are known to impact firn microstructure and bubble close-off characteristics, which in turn impacts gas records of $\delta O_2/N_2$ and total air content (Bender, 2002; Raynaud et al., 2007). Since $^{86}Kr_{xs}$ is linked to the dispersivity of deep firn, it seems probable that insolation has a direct impact on $^{86}Kr_{xs}$ also via the firn microstructure. We will revisit this issue in our interpretation of the WDC $^{86}Kr_{xs}$ record (Section*

*5). Overall, we anticipate $^{86}Kr_{xs}$ to be a qualitative proxy for synoptic variability, yet want to caution against quantitative interpretation based on the spatial regression slope."*

(5) In section 1.2, the authors discuss several processes that alter the isotope ratios, such as gravitational settling and thermal diffusion, advection, convection and dispersive mixing. The latter three they state do not distinguish between isotopologues. This is correct but they do lead to a disruption of the maybe already established isotope equilibrium through molecular (gravitational and/or thermal) diffusion, which requires time to be re-established. This is the starting point of their definition of the 86Krex40 proxy. However, there are several processes that can and will affect this proxy as nicely discussed in sections 1.2 and 3.3. This is of course a weakness of this proxy despite the author's transparent writing. For instance, they state that the major influence on 86Kr ex40 comes from pressure variations at the surface, but what about the pressure variations from the gas close-off process? Pressure variations may be weak but gas velocities of expelled molecules in the tiny channels at close-off depths might be very high and could lead to significant alterations in the gas compositions.

The reviewer is correct that pore closure from the firn densification process will drive an upward (macroscopic) air flux that will contribute to dispersive mixing throughout the firn column. To our knowledge, two previous studies have addressed this point. Both conclude that this effect is negligible compared to the barometric pumping driven by weather systems.

The first study is Schwander et al. 1988:

*"The decreasing porosity during firnification also leads to an air flow in the firn. When the firn is compacted, air is expelled from the open-pore volume. This leads to an upward movement of the air relative to the firn. The corresponding mean air velocity is of the order of the snow-accumulation rate, which is generally less than $10^{-4}$ mm/s. In the case of firn without melt layers (uniform upward flow relative to the firn at a given depth level), the flow-dependent part of the diffusivity is again negligible."*

The second study is Buizert and Severinghaus 2016:

*"Another source of macroscopic air movement in deep firn is the gradual closure of the pore space by the densification process, which leads to an upward air flow relative to the firn matrix (Rommelaere et al., 1997). The velocity of this (accumulation-rate-dependent) back flux is of the order of $10^{-9}$ to $10^{-8}$ m s$^{-1}$, and is clearly negligible in magnitude compared to the barometrically driven flow."*

To address this issue, we added the following sentence to section 1.2:

*"The upward air flow due to gradual pore closure is orders of magnitude smaller than the flows driven via barometric pumping, and neglected here."*

Minor points:

L144: it would be worthwhile to explain why this definition is less sensitive to thermal diffusion (give corresponding reference). Yet,

We changed to: *"The $^{86}Kr_{xs40}$ definition is preferred, because per unit mass difference $\delta^{40}Ar$ is less sensitive to thermal fractionation than $\delta^{15}N$ is (Grachev and Severinghaus, 2003a; 2003b)"*

L144ff: What about close-off fractionation? We know that Ne, He, will expelled during close-off. Therefore, large molecules such as N2 will be less affected than O2, Ar, etc. Kr is obviously between N2 and O2. Especially, Ar will be subject of expelling. In this regard, the second definition with δ15N would be preferred.

Out of the three gases (Ar, Kr, N$_2$) Ar is the only one that is impacted by close-off size fractionation. This definitely impacts the $\delta$Ar/N$_2$ ratios which are negative (Fig. A1B). Fortunately, $\delta^{40}$Ar is impacted only weakly by gas loss during close-off fractionation. We discuss this impact in appendix A1.

Note that both definitions are impacted by the $\delta^{40}$Ar gas loss correction, as $\delta^{40}$Ar is also needed for the thermal correction.

The observations make it clear that the $^{86}$Kr$_{xs40}$ definition has less scatter than the $^{86}$Kr$_{xs15}$ definition – compare Figs. A3 and A4.

L225ff: I would prefer the gas splitting. As it can be tested by many gas species measurements in contrast to the different ice core samples. There only replications can help.

Yes, so do we. It is more time consuming, though. In future work we plan to only use the gas splitting approach.

L239f How extended is this bubble-clathrate zone as the signals are extremely small. What was the criteria for the given number in the depth or time range.

We based our choice of the BCTZ depth/age range on observed positive anomalies in dO2/N2 in WDC ice, which occurred between 1000 and 1600 m depth.

L242ff "Some of the EDC samples analyzed had clear evidence of drill liquid contamination, which acts to artefactually lower 86 Kr ex40; the late Holocene data used here were not flagged for drill liquid contamination." Give a reference for this statement.

We added a reference to (Baggenstos et al., 2019). We did not make the determination of drill liquid contamination, but relied on the original study for this observation. We now also explain that d86Kr excess is lowered via isobaric interference on mass 82.

L247 22 per meg /permil: I do not understand this, error propagation leads to a higher combined uncertainty and not a lower!, this does not make sense or do I miss something here?

Good question. This depends on the value of the denominator, and for the values given in the paper we assume a $\delta^{40}$Ar of around 1.2 permil that is typical for WAIS Divide.

Starting from the definition of Eq (2), the uncertainty in the numerator is effectively equal to the uncertainty of the $\delta^{86}$Kr measurement. Because the value of the denominator is typically greater than 1, the uncertainty of the $^{86}$Kr$_{xs}$ appears smaller than that of $\delta^{86}$Kr (in the WDC example, 26/1.2 = 22). Of course in a relative sense the $^{86}$Kr$_{xs}$ error is much greater than the $\delta^{86}$Kr error.

We now clarify this in the text: *"Via standard error propagation, this results in a ~ 22 per meg ‰$^{-1}$ (2$\sigma$) analytical uncertainty for both $^{86}$Kr$_{xs40}$ and $^{86}$Kr$_{xs15}$ at a site like WDC where $\delta^{40}$Ar ≈ 1.2 ‰."*

L248 this would be necessary. At least you can split a 1600 g samples in two sub-samples

Yes, this would be a good thing to try in future studies. For traditional ice cores it is challenging to get a 1600 g sample, however, as we typically do not have access to the full core but only the gas piece. It would further lead to very skinny long samples with a lot of exposed surface areas, which is not ideal as this may result in gas loss during pumping. For blue ice sites (Taylor glacier, Allan Hills) we tend to have much larger samples available, yet at such sites the orientation of the stratigraphy is poorly known and true depth replicates are rarely true age-replicates. We will consider this for future work.

L253 BP also denotes Before Present, consider changing it.

Good idea. We changed it throughout to BRP.

L257ff Are there additional indications that melting has occurred, for instance from water isotopes or changed greenhouse gas concentrations?

Possibly. We have not made such measurements. It would presumably result in elevated $CH_4$ and $CO_2$. We are unsure what the impact on water isotopes would be. In the absence of the greenhouse gas measurements we rely on the noble gases, which we believe are a sensitive indicator for melt.

L268 Can you specify what modern climate means (time range)

In this statement we rely on the ERA interim reanalysis dataset we used, which is from 1979-2017. We now specify this:

*"pressure variability in the modern climate (here: 1979-2017 CE)."*

L274 How can a daily variable be compared to a decadal variable (86Kr ex40)?

In all our analyses we compare Kr-86 excess to the multi-decadal average $\Phi$ (1979-2017 CE). We now specify this more clearly:

*"In Fig. 3A we plot the site mean 86Krxs40 (with $\pm 1\sigma$ error bars) as a function of $\Phi$ (averaged over full 1979-2017 period)"*

L278f If such a calibration is made, it should be done on firn air samples as they are not smoothed by the process of gas enclosure. Have you tried to do this?

We do have very limited firn air data for $\delta^{86}Kr$, and we do not have access to stored firn air from a wide range of climatic conditions. So unfortunately, we cannot perform this analysis on firn air. Furthermore, we find that firn air $^{86}Kr_{xs}$ does not match the values measured in mature ice samples. We have no clear explanation for this mismatch. We discuss this in section 3.3:

*"Measurements on firn air samples, where available, suggest a smaller 86Krxs anomaly in firn air than found in ice core samples from the same site. We attribute this in part to a seasonal bias that is introduced by the fact that firn air sampling always takes place during the summer months, whereas the synoptic variability that drives the Kr-86 excess anomalies is largest during the winter (Fig. 2C); consequently, firn air observations are biased towards weaker 86Krxs. Further, in the deep firn where 86Krxs anomalies are largest, firn air pumping may not yield a representative air sample, but rather be biased towards the well-connected porosity at the expense of poorly-connected cul-de-sac-like pore clusters. Since barometric pumping ventilates this well-connected porespace with low-86Krxs air from shallower depths, the firn air sampling may not capture a representative 86Krxs value of the full firn air*

*content. These explanations are all somewhat speculative, and a definitive understanding of the firn-ice differences is lacking at this stage."*

L288f This is only the case when the argon correction for gas loss has been made correctly.

They would be identical for any value of $\varepsilon_{40}$. But the result indeed relies on the argon gas loss correction. We believe we have been forthright about this, and dedicated several figures and an appendix to this correction (Figs 3B, A3, A4).

L299ff Refer to Figure 3.

We added a reference to Fig. 3A as suggested.

L304ff This is disconnected, the link is not clear. Further explanation is needed here.

Thanks for this feedback, it is indeed probably not very clear to readers unfamiliar with the cited Koutavas paper. We have instead removed this sentence, as it is not important to the paper (which is long enough as it is!).

L310ff This is indeed a critical point.

Yes, it really is. We believe all the points in section 3.3 are critical, which is why we try to be very cautious in our writing and be clear that the results are tentative at this point.

L332f This is again a critical point as I can imagine that the firn structure acts as a column retarding the gas species differently. It would be worthwhile to do such experiments. Maybe you find a corresponding reference?

Agreed. We are considering writing a follow-up proposal to do such experiments. For now we can only speculate, unfortunately.

L346ff This is also important. Elemental ratio should be in line with isotope ratios. Yet it points indeed to a difference in diffusion coefficient ratio of real and lab conditions. Column effect (adsorption/desorption)

Agreed. We edited this paragraph in revision to include the adsorption effect:

*"Firn models predict that the gravitational disequilibrium effect in elemental ratios (such as Kr/Ar) should be proportional to that in isotopic ratios. However, the observations suggest that the former is usually smaller than would be expected from the latter. As before, adsorption of Kr onto firn grain surfaces may contribute to the observed discrepancy, and laboratory tests of this process are called for. Further, the impacts of gas loss are greater on elemental ratios than on the isotopic ratios which may contribute also. Including measurements of xenon isotopes and elemental ratios in future measurement campaigns may be able to provide additional constraints to better understand this discrepancy."*

L361f if one argues that the diffusion of noble gases may be retarded in the firn column, one should consider this effect also for the thermal corrections. These, however is based on δ15 N and δ40 Ar measurements.

The estimated diffusivities of $^{15}$N and Ar are very similar (Buizert et al., 2012), so therefore we ignore this to first order. The effect of the firn column on retarding these via adsorption are not well known. Ar

has a higher solubility, and therefore is likely to have a stronger adsorption effect. However, the adsorption for both these gases is estimated to be small compared to Kr and Xe though.

L368ff this is also a very critical correction as obliviously ΔT varies considerably from site to site without a clear understanding why this is the case.

Yes, we agree. The reviewer does not seem to be asking for a correction or response here.

L378 give a reference for (1)

We added the reference to Baggenstos et al., (2019)

L379 (3) yes, this indicates the large uncertainty of this correction. Yet, Figure 3B is quite convincing as a counter-argument. See also main comments above.

Agreed. We responded to the main comments above, no further response seems needed here.

L 410 How has the elevation changed over the course of the investigated period? And how relevant are these changes?

This is not well known, and we need to rely on models here. For WDC, Golledge et al. (2014) simulate an LGM elevation of around 300 m higher than at present. Via the regression slope, this would result in a 10 per meg ‰$^{-1}$ change in $^{86}Kr_{xs}$. This is within the uncertainty of our measurements, and smaller than the temporal signals we interpret.

L416 ...(by limiting ...), not clear, needs further explanation

We clarified this statement to read: *"via its topographic influence on the position of storm tracks"*

L429f ...we anticipate 86Kr ex40 to be a qualitative proxy for synoptic variability... this is indeed a good point as the used calibration is standing on weak grounds.

The reviewer does not seem to be asking for a correction or response here.

L456ff Why have you only investigated Antarctic sites and not Greenland locations? This would proof that different locations on Earth would be similarly influenced. There is GISP2. What would be a good choice for additional stations in Greenland?

A map of synoptic variability in Greenland is given in figure 8 of Buizert and Severinghaus (2016). It is clear that Greenland has only a small range of barometric variability compared to Antarctica. The main coring sites in the interior all have a comparable level of pressure variability, making them uninteresting for the spatial calibration. Cores near the margin, such as the Renland core, have greater pressure variability, yet nothing near the levels seen in coastal Antarctica. Greenland coastal sites like Renland suffer from summer melting, however, which impacts the Kr and Xe inventory of the ice strongly. For this reason we decided to use only a single Greenland core in this study. In the future it would be interesting to obtain a long-term record of Kr-86 excess from Greenland summit.

L549f or is there indeed a higher variability present. How do you explain or underline that less care has been taken for these later campaigns?

These measurement campaigns were aimed at obtaining a $\delta Kr/N_2$ record for reconstructing mean ocean temperature, and Kr-86 excess was a by-product of those measurements. For these reasons less care was given to the isotopes than desired. Perhaps the experimenters were looking to complete the project on a short time schedule.

L575f Campaign 3 data shows quite a large scatter.

- We have added this caveat to the text: *"The trend in campaign 3 is less robust due to the greater scatter in the data"*

L600ff this might be tackled with measurements in Greenland compared to those in Antarctica

This is a great idea for future studies.

---

## Author Comment (AC3)

**Comments by Reviewer #2**

We want to thank Reviewer #2 for their thoughtful and constructive comments that have improved the manuscript. Our responses below. We wrote our responses in the form of proposed changes to the text that we would make in a potential revised manuscript.

This manuscript presents a compilation of isotopic composition of krypton which has been obtained over the past years in different ice cores from Greenland and Antarctica.

Using the present-day understanding of the drivers of gas repartition in the firn as well as correlation with ERA products, the authors propose that the 86Kr_excess can be used for synoptic activity.

The manuscript is generally well written and well documented. The authors also explained in details the numerous limitations associated with this interpretation both in section 3.3 as well as in the supplements.

Because of the limitations in the interpretation of the 86Kr_excess, the authors should be more cautious in the proposed interpretation. While the scientists in the filed of ice cores will get the limitations and use the results with caution, it may be different for people who do not understand the complexity of processes affecting air elementar and isotopic repartition in the firn. I thus suggest to modify the abstract and the conclusion to insist on the speculative interpretation of the 86Kr_excess and on the additionnal measurements to be done to better quantify the effect of gas loss, thermal diffusion (including rectifier effect) and possible existence of a convective zone.

We of course fully agree with the reviewer on this point, and we tried to be very careful in our wording throughout. Reviewer #1 also commented we were careful in our interpretation. We see the risk that the reviewer highlights, as people tend to plot records that support their argument without always including all the caveats from the source publication. However, one can never be too cautious, and so in response to this request, we have made the following changes:

In the abstract we added:

*"Limited scientific understanding of the firn physics and potential biases of $^{86}Kr_{xs}$ require caution in interpreting this proxy at present."*

We further replaced the word "*tentative*" with "*speculative*".

In the conclusion, we added:

*"Due to these limitations, we caution that any interpretation of temporal $^{86}Kr_{xs}$ changes remains speculative at present."*

146 : it is strange to refer to Fig 6 here. Moreover, it is strange to prefer one or the other since it is shown later that gas loss and thermal effect are the most important corrections to take into account (d40Ar being sensitive to gas loss and d15N being the most sensitive to thermal fractionation). There is no obvious reason to prefer one notation compared to another.

We have removed the reference to Fig. 6 in this place in the paper.

For most cores drilled in the past 2 decades where ice samples have been stored under cold conditions, we expect that gas loss correction to be the smaller correction with the smaller uncertainty. Furthermore, the observations make it clear that the $^{86}Kr_{xs40}$ definition has less scatter than the $^{86}Kr_{xs15}$ definition – compare Figs. A3 and A4. For these reasons we prefer the $^{86}Kr_{xs40}$ definition. We have provided plots for each of the definitions, both for the WDC record and for the spatial calibration study. Therefore, readers can compare both definitions and derive their own conclusions for the comparison.

From l. 223 : the preparartion of the samples is different for DE08-OH than for the other samples. May this explain the different slopes associated with gas loss in figure A1-B.

Yes, possibly. This is a good suggestion that we had not considered. For the DE08-OH samples, the dO2/N2 and dAr/N2 were measured on smaller samples (larger surface-to-volume-ratio), which may be more affected by gas loss during pumping on the samples. We added this to the supplement A1:

*"The DE08-OH samples were also analyzed differently from those at other sites, with $\delta O_2/N_2$ and $\delta Ar/N_2$ measurements performed on a separate smaller ice piece (see section 2.2); the greater surface-to-volume ratio of such small samples may result in greater gas fractionation while evacuating the sample flasks in the laboratory."*

241 : How are the samples flagged for drill liquid contamination ? How is it possible to detect the drill fluid contamination ? From which measurements ?

Here we rely on Baggenstos et al. (2019), where the EDC data originate. The original study flagged the drill liquid contaminations, which can be observed in the IRMS analysis. Baggenstos et al. report in the supplementary information that drilling fluid contamination causes isobaric interference on mass 29 and 82, thereby impacting measurements of both $\delta^{15}N$ and $\delta^{86}Kr$. We have now added this to the manuscript.

245 – 247 : Can you explain the error propagation explaining why the 2 sigma is larger for 86Kr than for 86Kr_excess ?

This is a good point also brought up by reviewer #1. We copied the response to reviewer #1 here:

Good question. This depends on the value of the denominator, and for the values given in the paper we assume a $\delta^{40}Ar$ of around 1.2 permil that is typical for WAIS Divide.

Starting from the definition of Eq (2), the uncertainty in the numerator is effectively equal to the uncertainty of the $\delta^{86}Kr$ measurement. Because the value of the denominator is typically greater than 1, the uncertainty of the $^{86}Kr_{xs}$ appears smaller than that of $\delta^{86}Kr$ (in the WDC example, 26/1.2 = 22). Of course in a relative sense the $^{86}Kr_{xs}$ error is much greater than the $\delta^{86}Kr$ error.

We now clarify this in the text: *"Via standard error propagation, this results in a ~ 22 per meg ‰$^{-1}$ (2$\sigma$) analytical uncertainty for both $^{86}Kr_{xs40}$ and $^{86}Kr_{xs15}$ at a site like WDC where $\delta^{40}Ar \approx 1.2$ ‰."*

253 : I do not see why it is useful to present these data to remove them immediatly after. In this case, the 86_Kr data from EDC samples affected by drill fluid should also be displayed with an explanation on how they were discarded.

Yes, that is an interesting point. The data from these two sites (JRI and BRP) are not really displayed however, and only shown only in Figure A1A. The full isotope measurements are archived online for others to use, however. The samples from these two sites were provided and shipped by our international collaborators for the sole purpose of this calibration study, as we thought the Antarctic Peninsula was an important site to include. Therefore, we feel an obligation to report these failed attempts. It further provides the important lessons that melt should be avoided for Kr-86 excess (which is no surprise given the high solubility of Kr), and that refrozen meltwater can be present in the absence of visible melt features (which has not been reported before to our knowledge).

The samples from EDC were not measured for the purpose of this calibration study, however, and were previously published by Baggenstos et al. (2019). The drill fluid flagging was not done by our team, we simply rely on the original study that had flagged these samples. Therefore the choice to discard them was not ours, and justification for this choice is given in the original study (Baggenstos et al., 2019).

Section 3.1 : The Phi parameter exhibits strong seasonal and interannual variabilities and I do not understand how this variability is taken into account in the « calibration » of the 86Kr_excess. Such sensitivity should be studied or implemented in Figure 3 since this is crucial for the interpretation of the 86Kr_excess proposed here.

The seasonal and interannual variability is discussed to give the reader a better understanding of the nature of synoptic variability in Antarctica. However, neither impacts the calibration study, where we compare Kr-86 excess to the multi-decadal average $\Phi$ (1979-2017 CE). We now specify this more clearly:

*"In Fig. 3A we plot the site mean 86Krxs40 (with ±1$\sigma$ error bars) as a function of $\Phi$ (averaged over full 1979-2017 period)"*

The samples each represent multiple years – both because of the wide age distribution of firn air, and because the large samples needed typically span multiple annual layers.

Section 3.3 : this section is interesting in providing the limitations of the interpretation of 86Kr_excess and strongly suggest that further study should be performed for a robust interpretation such as firn air pumping study at different site with a correct determination of the thermal gradient (it is really surprising to find such temperature gradient at DE08 and EDC) + analyses of ice not affected by gas loss, etc… this is the reason why the authors should be much more cautious in their conclusions and better suggest concrete perspectives on how to progress with such proxy if it is reallt promising. Actually, the concluding paragraph of section 3.3 should also be summarized in both the abstract and conclusion of the manuscript to clearly state the limit of this interpretation which is now speculative.

We agree. We are the first to admit that this proxy has challenges at present, which is why we devote an entire section of our paper to it. In response to the first reviewer #2 comment, we already added additional notes of caution to the abstract and conclusions, and specifically note that the interpretation remains speculative (using the reviewer's choice of words).

We further added to the abstract: *"A list of suggested future studies is provided."*

And to the conclusions: *"A full list of suggested follow-up studies is given in section 3.3"*

In section 4, I feel that a discussion on the seasonal variability and its possible impact is missing.

The gas age distribution at the base of the firn has a width of several years, and therefore Kr-86 excess reflects the time-averaged barometric variability over several years. For this reason, we only investigate the annual-mean pressure variance. Including an additional analysis of seasonal variability would not change any of our conclusions, and add to an already overly long article.

Section 5 : I understand that the authors do their best with the poor data quality but it would be nice to comment on the strong scattering for the data at « present-day » ? Can this scattering be used to estimate the uncertainty as the authors mention that « no true replicate to assess the reproducibility » …

This is an interesting point, and it is true that there is substantial scatter in the records. We are unclear what this scatter represents, at present. As the reviewer suggests, there may indeed be a contribution from analytical uncertainty which is not captured in our estimated precision. However, climate patterns such as the Pacific Decadal Oscillation definitely impact storminess at WDC in reanalysis data, which suggests some of this scatter probably reflects variations in WDC storminess at decadal or inter-annual time scales. Last, the DE08-OH data suggest there may be cm-scale variations in Kr-86 excess that we attribute to layering in firn microstructural properties. In our sampling we may not be fully averaging out this cm-scale variability, contributing to more scatter. We added this to the text in Section 5:

*"The scatter in the late Holocene WDC $^{86}Kr_{xs}$ data exceeds the stated analytical precision. Potential explanations include (1) an underestimation of the true analytical precision; (2) interannual to decadal variations in storminess at WDC; and (3) aliasing of cm-scale variations in ice core $^{86}Kr_{xs}$ linked to layering in firn microstructural properties."*

20 and 21 : the discussion is quite long for such speculative interpretation. I would suggest shorten it to stay on the safe side of the interpretation.

In response to this comment we shortened the interpretation section by removing the discussion about the split jet.

725 : I am not sure that the authors really « calibrate » the proxy – let's say that this is a first proposition of interpretation. A calibration would require more dedicated studies as mentionned in the concluding paragraphe of section 3.3.

We removed the word "calibrate" here.

Figure 3 : What is the origin of the uncertainty bars for the different sites ? Do the sites with more data have more scattering hence a larger uncertainty bar ? It would be useful to mention the number of points used for each sites in this calibration and how the error bar is calculated. A table may be useful to exactly describe the number of samples for each site, depth range, conditions of storage, etc…

The number of samples for each site in the calibration study is listed in Table 1. The number of samples and nature of the error bars is explained in the updated figure caption:

*"the error bars denote the ±1$\sigma$ standard deviation between samples (uncertainty in corrections and measurements not included). The number of samples at each site is listed in Table 1."*

A more complete description of the sample characteristics (precise depths etc.) is provided by the data tables archived online.

Figure A2 : The displayed results show very depleted samples in dO2/N2 and dAr/N2 – are these results really relevant for this paper ? What is the origin of these samples ? core top ? Bottom ice ?

The relevance to this paper is that we use the Byrd data to estimate the impact of gas loss on $\delta^{40}$Ar. This is described in Appendix A1. We now specify this more clearly in the figure caption:

*"Argon isotopic enrichment due to gas loss in the Byrd core used to determine the $\delta^{40}$Ar gas loss correction (appendix A1)."*

Figure B1 shows that there may be a large scattering with depth of 86Kr-excess. I am sure that this is taken into account in this study but it would be nice to explain a little bit more how it is done (also for the other cores). Probably again a table explaining the number of samples considered for present-day for each core, the depth range and individual values would help.

The number of samples is listed in Table 1. We added the following statement to account for the other sites:

*"At all other sites analyzed here, the sample length exceeds the annual layer thickness; this will remove some, but not all, of the effects of the sub-annual variations."*

Citation: https://doi.org/10.5194/cp-2022-65-RC2

---

## Author Response (AR2)

Dear Hubertus,

Thank you for your detailed review or our revised manuscript. We have made the changes you requested – in most cases directly following your suggestions. Please find detailed comments below.

All the best,

Christo, on behalf of the authors

additional points:

The wording of the abstract does not fully reflect the tentative character of your conclusions and the detailed discussion of the proxy in the manuscript (see also several comments by referee #1). I would suggest the following wording changes to the abstract:

line 36 of your track changes version: "The 86Krxs may therefore reflect the time averaged..., but may not record individual synoptic events."
line 43 of your track changes version: "We show that Antarctic 86Krxs appears to be linked to the ..."
line 46 of your track changes version: "... from the WAIS Divide ice core. Based on the empirical spatial correlation of synoptic activity and 86Krxs at various Antarctic sites, we interpret this record to show that synoptic Activity is slightly below..."

We have made these suggested changes, with minor changes from the suggested wording for the first point.

line 121 of your track changes version: "... and T temperature in Kelvin."
Done

The discussion of the effusion effect in line 132 is a bit unconnected to the rest of the text and should be backed up at best by a calculation of the air flow or at least the respective references should be provided. I would suggest to write at line 132 of your track changes version: "Note that also a upward air movement exists in the firn column relative to the overall downward advection of the ice, which is caused by the slow reduction of porosity with depth. However, this upward air flow..."
Following this sentence I would recommend to give a back of the envelope calculation here for the speeds of the air movements.

The calculations came from Buizert and Severinghaus (2016, page 2103). That paper had been cited on the previous line – which is why we did not repeat the citation again. We now have expanded this following your suggestion:

"Note that also an upward air movement exists in the firn column relative to the overall downward advection of the ice, which is caused by the slow reduction of porosity with depth (Rommelaere et al., 1997). This upward air flow due to gradual pore closure (around $10^{-9}$ to $10^{-8}$ m s$^{-1}$) is orders of magnitude smaller than the flows driven via barometric pumping (around $10^{-6}$ m s$^{-1}$), and therefore neglected here (Buizert and Severinghaus, 2016). "

line 290 of your track changes version: Please provide a reference for the timescale of pressure equilibration or some argumentation why it is about one hour

That calculation is again from Buizert and Severinghaus (2016). We added the citation.

line 389 of your track changes version: "...in the absence of a gradient DeltaT in mean annual temperature (Morgan et al., 2022)."

Agreed. This is an important addition.

line 395 of your track changes version. "... an unexpected positive DeltaT..."

Done

line 558 ff of your track changes version: I assume you use the mean of the next neighbor residuals to assess the significance of the offset between campaign 1 and 2. Each residual contains the analytical uncertainty of 2 individual measurements (uncertainty is larger), but at the same time you need to show that the standard error of the mean of all residuals is not significantly from zero (uncertainty becomes smaller). Please expand the discussion of the uncertainty in this paragraph accordingly.

The measurement offsets between the five campaigns are easily visible by eye, and we just wanted to provide a simple way to estimate these offsets. We did not explain very carefully how we performed the linear interpolation, so we added some clarification to this section of the manuscript.

line 627 ff of your track changes version: This paragraph is somewhat disconnected to the rest. Please extend a bit to explain.

For more context, we added:

"Understanding the cause of this relatively high scatter in the $^{86}Kr_{xs}$ records will require more work, in particular the measurements of several high resolution $^{86}Kr_{xs}$ records in various sectors of Antarctica."

line 694 of your track changes version. "... a statistically significant impact at WDC..."

Thanks. Corrected.

libe 768 of your track changes version: I would delete ", demonstrating the validity of the new proxy."

Agreed. The reader can decide for themselves whether the proxy is valid or not.

caption Figure 3 line 1251 of your track changes version: "15N excess"
Done

Figure 8: Are the data points in this figure corrected for gas loss and thermal fractionation or not? Please add this information to the caption. Moreover the spline in this figure shows a peak at around 17 kyr BP which is not backed up by a data point in campaign 1 or 2 (i.e. the peak in the spline is higher than all the data points). Please double check or explain, why this peak appears.

There is no Figure 8, and we are not sure whether you refer to fig 6 or 7 here.

Thanks for catching the difference between the spline and data! We had to dive back into the code to find the origin of this issue, which was an error in our previous submission. We had inconsistently applied the gas loss correction to the data and the spline.

For campaigns 4 and 5 we do not have $\delta O_2/N_2$ or $\delta Ar/N_2$ data, which makes it impossible to apply the gas loss correction. To compare campaigns 1-5, we had NOT applied the gas loss correction to any of the data in the old figure 6. However, the gas loss correction WAS being applied to the spline (which was based on campaigns 1 and 2 only).

We have made the following changes:

- We have made a 3$^{rd}$ order polynomial fit to the WDC $\delta O_2/N_2$ or $\delta Ar/N_2$ data from campaigns 1-3, which allows us to make a systematic gas loss correction to all campaigns.
- We have added this fit to figure A5.
- We have updated figures 6 and 7 with data and splines consistently corrected for gas loss and thermal fractionation

We added the following text to appendix A1 on the gas loss correction:

"In order to provide a consistent gas loss correction to the five measurement campaigns, including campaigns 4 and 5 for which no $\delta O_2/N_2$ or $\delta Ar/N_2$ data are available, we fit a third-order polynomial to all available gravitationally-corrected WDC $\delta O_2/N_2$ - $\delta Ar/N_2$ data (Fig. A5A). We can then calculate the expected WDC $\delta O_2/N_2$ - $\delta Ar/N_2$ at any given age, also in the absence of $\delta O_2/N_2$ and $\delta Ar/N_2$ data. For consistency, we use this correction method for all data seen in Fig. 6. Note that the WDC $\delta O_2/N_2$ - $\delta Ar/N_2$ values are small for all ages, and that therefore the gas loss correction is small for this site."